# Correlative single-molecule and structured illumination microscopy of fast dynamics at the plasma membrane

Hauke Winkelmann [1], Christian P. Richter[1], Jasper Eising[1], Jacob Piehler [1,2] & Rainer Kurre [1,2,3]

Total internal reflection fluorescence (TIRF) microscopy offers powerful means to uncover the functional organization of proteins in the plasma membrane with very high spatial and temporal resolution. Traditional TIRF illumination, however, shows a Gaussian intensity profile, which is typically deteriorated by overlaying interference fringes hampering precise quantification of intensities—an important requisite for quantitative analyses in single-molecule localization microscopy (SMLM). Here, we combine flat-field illumination by using a standard πShaper with multi-angular TIR illumination by incorporating a spatial light modulator compatible with fast super-resolution structured illumination microscopy (SIM). This distinct combination enables quantitative multi-color SMLM with a highly homogenous illumination. By using a dual camera setup with optimized image splitting optics, we achieve a versatile combination of SMLM and SIM with up to three channels. We deploy this setup for establishing robust detection of receptor stoichiometries based on single-molecule intensity analysis and single-molecule Förster resonance energy transfer (smFRET). Homogeneous illumination furthermore enables long-term tracking and localization microscopy (TALM) of cell surface receptors identifying spatial heterogeneity of mobility and accessibility in the plasma membrane. By combination of TALM and SIM, spatially and molecularly heterogenous diffusion properties can be correlated with nanoscale cytoskeletal organization and dynamics.

Resolving the dynamic functional organization of biomolecules in the cell remains a key challenge for a fundamental understanding of cell biology. Many cellular processes occur in the context of lipid membranes that further enhance the complexity by providing distinct, spatially heterogeneous environments[1]. As a paradigm for such complexity, the plasma membrane (PM) exhibits dynamic mosaicity at nanoscale, which is determined by an intricate interplay of membrane protein and lipid interactions in the context of underlying cortical actin cytoskeleton[2–7]. The function of membrane proteins in the PM, e.g., transmembrane signaling complexes, are highly regulated by such distinct and spatially heterogeneous membrane properties[8–13]. Thus, the key role of dimerization in the activation of cytokine receptors has been recognized, yet the exact principles driving assembly of signaling complexes has remained controversially debated[14–18]. Similar questions have emerged for other types of PM receptors including G-protein coupled receptors (GPCRs)[19,20].

[1]Division of Biophysics, Department of Biology/Chemistry, Osnabrück University, Barbarastraße 11, D-49076 Osnabrück, Germany. [2]Center for Cellular Nanoanalytics, Department of Biology/Chemistry, Osnabrück University, Barbarastraße 11, D-49076 Osnabrück, Germany. [3]Integrated Bioimaging Facility iBiOs, Department of Biology/Chemistry, Osnabrück University, Barbarastraße 11, D-49076 Osnabrück, Germany. ✉e-mail: piehler@uos.de; rainer.kurre@uos.de

Interactions of transmembrane receptors are often highly transient and membrane context-dependent, causing spatial, highly dynamic heterogeneity at nanoscale level. Furthermore, receptor diffusion in the PM is intricately convolved with processes related to membrane trafficking, with the endocytosis of signaling complexes emerging as a multifaceted regulator of downstream effector activation[21–24].

Resolving such complex and dynamic functional organization in cells remains highly challenging. Modern far-field fluorescence microscopy techniques offer spatial resolution far beyond the diffraction limit of light, e.g., by stimulated emission depletion (STED) or minimal photon fluxes (MINFLUX)[25], which also can provide information on diffusion properties with ultrahigh spatiotemporal resolution[25–28]. However, these techniques do not readily allow for simultaneously imaging nanoscale topography and molecular dynamics. Imaging with resolution down to molecular dimensions can likewise be achieved by single-molecule localization microscopy (SMLM) techniques[29]. Live cell SMLM also opens versatile opportunities for visualizing mobility and interactions with very high spatial and temporal resolution[19,30–34]. Detecting and characterizing the formation of protein complexes by SMLM can be based on intensity analysis[35,36], dual or multiple color co-localization/co-tracking[37–40] or single-molecule Förster resonance energy transfer (smFRET)[41,42]. Total internal reflection fluorescence (TIRF) microscopy offers powerful means for single-molecule imaging of proteins in live cells, in particular for imaging the PM with utmost axial resolution. By tracking individual molecules, spatiotemporal analysis of trajectories reveals single protein dynamics and interactions involved in important cellular processes such as signaling, endocytosis or trafficking. A traditional objective-based TIRF microscope is based on focusing a single Gaussian laser beam into the TIR ring of the objective's back focal plane. This results in a totally internally reflected laser beam at the glass/sample medium interface and a 100–200 nm thin evanescent field for fluorescence excitation directly above the coverslip. The evanescent field of such a microscope shows a Gaussian intensity profile which is typically deteriorated by overlaying interference fringes, which hampers a precise quantification of single-molecule intensities—an important requisite, e.g., for stoichiometry or smFRET analyses. Flat-top TIRF illumination has been shown to tremendously enhance quantitative single-molecule imaging[43]. However, fast imaging of nanoscale structures is not possible by these approaches.

Here, we describe a microscope dedicated to rapidly switch between highly homogeneous and structured TIR illumination to enable simultaneous quantitative single-molecule and SIM imaging with high temporal resolution. To this end, we devised flat-field TIRF illumination based on a standard πShaper combined with multi-angular TIR illumination by incorporating a spatial light modulator (SLM). Using a class I cytokine receptor as a model system, we demonstrate robust detection of dimerization by single-molecule intensity analysis and smFRET. Homogeneous illumination furthermore allowed for long-term tracking and localization microscopy (TALM) of receptor dynamics in the plasma membrane uncovering striking heterogeneity in accessibility and diffusion properties. Combination of quantitative TALM and SIM enabled correlation of receptor mobility with cytoskeletal structures and the detection of endocytic uptake and trafficking.

## Results

### Homogenous multi-angular TIR illumination

We designed a microscope setup which enables quantitative multi-color single-molecule fluorescence microscopy (SMFM) and high-speed super-resolution SIM in living cells (Supplementary Fig. 1a). To achieve TIR illumination with utmost homogeneity, we introduced a πShaper into our setup which transforms a Gaussian laser beam into a flat-top intensity profile (Supplementary Fig. 2a, b)[43]. The π-shaped beam illuminates a spatial light modulator (SLM) which is conjugated to the final focus plane of the microscope's objective. We used a binary ferroelectric liquid crystal on silicon device (FLCOS-SLM) known for its very fast display update rates in the kHz regime. The SLM displays binary phase gratings and their first-order diffractions are imaged to the back focal plane of the objective (Fig. 1a). By changing the period of the phase gratings, one can easily switch illumination mode from total internal reflection (TIR) to grazing incidence (GI) and highly inclined and laminated optical sheet (HILO) illumination[44,45]. Homogenous multi-angular illumination during a single camera exposure is achieved by cycling through nine phase gratings with equidistant angles of orientation (0°, 60°, 120°) and three phase shifts (0, $2/3\pi$, $4/3\pi$) each. Multi-color TIRF microscopy at the same penetration depth is possible by cycling through different sets of phase gratings calculated for the design wavelength and penetration depth (Supplementary Fig. 1c, d). Exactly the same gratings were used for super-resolution SIM. For single-molecule tracking (SMT) and FRET, we used an EMCCD camera with maximum frame rate of 50 Hz at full resolution. SIM data with maximum frame rate of 216 Hz was acquired by an sCMOS camera. Real-time live cell imaging requires high-speed multi-color detection with low emission crosstalk. We therefore used two four-channel image splitters for SMFM and SIM and no slow mechanical filter wheels.

We first explored the capabilities of our multi-color flat-field SIM setup by imaging a dye solution containing fluorescein, Texas Red and ATTO 655, respectively. Each illumination mode (HILO, GI, TIR) can be sequentially switched by the SLM without the need of any further optical adjustments (Supplementary Movie 1). Alignment and focus were confirmed by a symmetric fluorescence signal from all six laser beams. Especially TIRF microscopy requires perfect alignment. We analyzed all accessible illumination modes based on a Gaussian versus a flat-top illumination profile (Supplementary Fig. 2c–f) and demonstrated the homogeneity of multi-channel TIR fluorescence by line plots for each channel (Fig. 1b, c). Having confirmed highly homogeneous TIRF microscopy with bulk fluorophores, we explored the performance at single-molecule level using transient binding of the fluorogenic substrate MaP555 to immobilized reHaloTags[46]. To this end, His-tagged reHaloTagF was immobilized on glass coverslips rendered biocompatible and functionalized with tris-(nitrilo triacetic acid) tris-NTA[47]. Time-lapse TIRF imaging enabled robust detection of individual MaP555 (Fig. 1d, Supplementary Movie 2). A maximum intensity projection from 4000 consecutive frames and the single-molecule localization map confirmed highly homogeneous excitation and detection at single-molecule level (Fig. 1e, f). The corresponding intensity histogram is dominated by a single Gaussian distribution (Fig. 1g) with a width in line with simulations of shot noise-limited PAINT experiments (Supplementary Fig. 3b–d). A quantitative comparison with single-beam TIRF microscopy (Gaussian versus flat-top) clearly demonstrates the striking benefits of our method (Supplementary Fig. 4). Potent high-resolution single-molecule localization across the entire field of view was confirmed by dual color DNA-PAINT imaging of DNA origamis (Supplementary Fig. 5a–e). Likewise, highly homogenous TIRF illumination imaging was observed upon imaging the plasma membrane of HeLa cells transiently expressing the C-terminal fragment of HRAS responsible for farnesylation and plasma membrane targeting fused to GFP (farnesyl-GFP, Supplementary Fig. 5f, g).

### Robust intensity-based quantification of receptor stoichiometries in living cells

Taking advantage of the highly homogeneous TIRF illumination, we explored intensity-based stoichiometry analysis of receptors in the plasma membrane of live cells. We have previously found that the thrombopoietin receptor (TpoR) is expressed at low cell surface

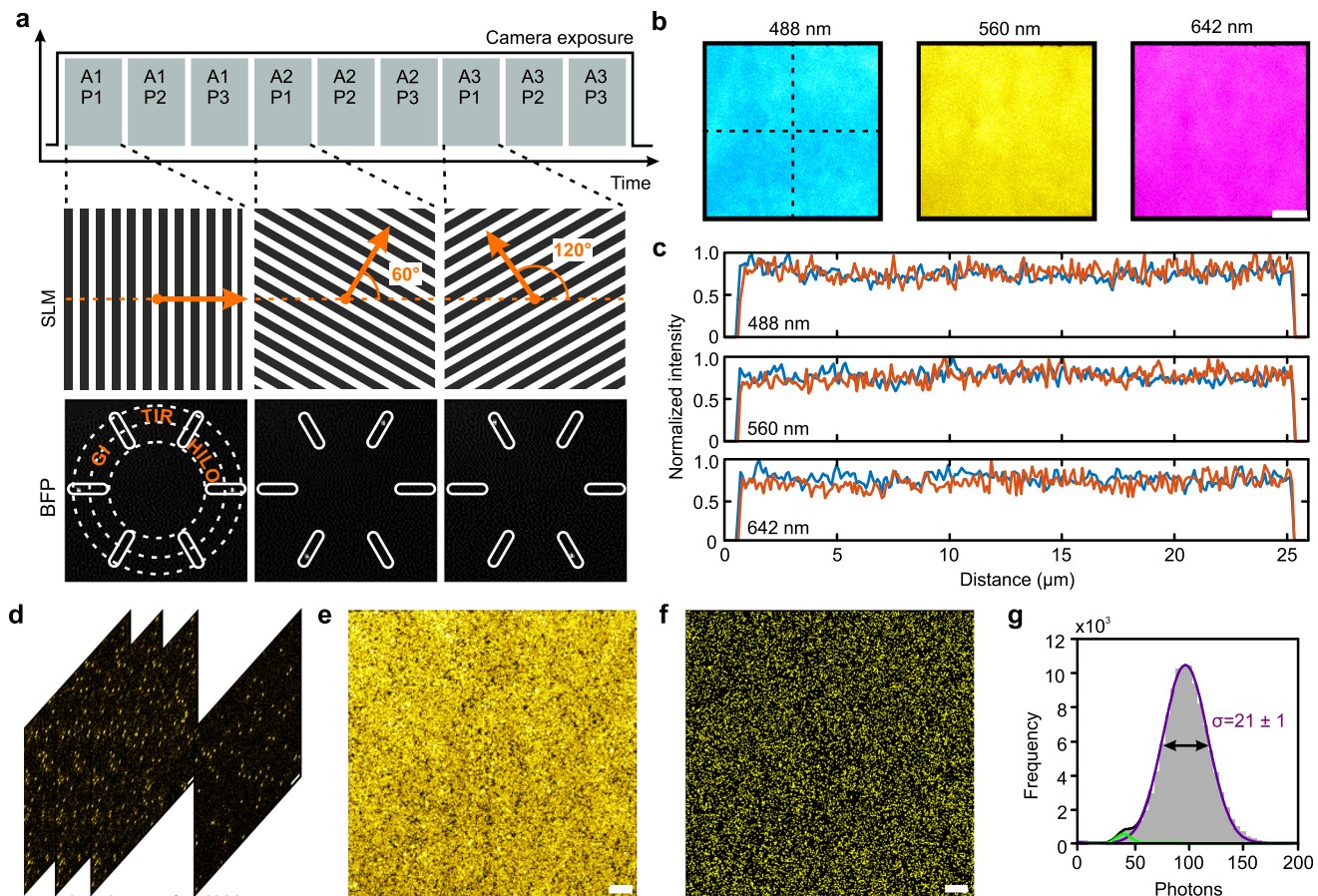

**Fig. 1 | Homogenous flat-field illumination by fast angle and phase switching of structured illumination patterns. a** Illustration of phase gratings displayed on SLM and switched within single camera frame acquisition (A: angle, P: phase). Binary phase gratings on SLM and corresponding filtered diffraction patterns after first-order mask at back focal plane (BFP) measured by inspection camera. TIR ring, position of grazing incidence (GI) and HILO illumination are indicated by dashed lines. Pattern switching and illumination is also illustrated by supplementary movie 1. **b** Homogeneous illumination by flat-field TIRF-SIM. TIRF images of dye solutions excited with different laser lines: 488 nm (fluorescein), 561 nm (Texas

Red) and 640 nm (ATTO 655) by cycling through all angles shown in **a** (scale bar, 5 μm). **c** Line profiles along dashed lines in **b** for all three channels (blue: horizontal; orange: vertical). **d–g** Single-molecule imaging of MaP555 transiently immobilized on a glass cover slide coated with reHaloTagF. Time-lapse TIRF imaging **d**, maximum intensity projection **e** and localized molecules **f** from 4000 consecutive frames. Scale bar: 2 μm. **g** Intensity histogram fitted by a bimodal Gaussian function (n = 1.11 × 10⁵ localizations). Standard deviation σ of main peak is indicated. Source data are provided as a Source Data file.

densities and is strictly monomeric in the absence of ligand[41], which makes it a very useful model system. To achieve well-defined, cell surface-selective labeling for quantitative stoichiometric analysis, we employed TpoR N-terminally fused to a non-fluorescent monomeric GFP-tag (mXFP) for labeling via anti-GFP nanobodies "enhancer" site-specifically conjugated with ATTO 643 (${}^{AT643}$EN, Fig. 2a)[39]. To control TpoR dimerization, an ALFA-tag[48] was inserted upstream of the mXFP-tag, which was crosslinked by a tandem anti-ALFA nanobody (tdAL-FAnb, Fig. 2b). HeLa cells transiently expressing ALFA-mXFP-TpoR were imaged in the presence of ${}^{AT643}$EN at video-rate (Supplementary Movie 3). Intensity analysis in the absence of the dimerizer yielded a highly homogeneous Gaussian intensity distribution (Fig. 2c, d) with a mean signal of ~400 photons/particle.

Upon dimerization of TpoR by addition of the crosslinker tdALFAnb, particles with correspondingly enhanced intensity could be discerned (Supplementary Movie 3, Fig. 2e). The intensity distribution analysis showed an additional peak at higher intensity. The intensity histogram could be fitted by a bimodal Gaussian distribution with maxima at ~400 and ~800 photons/particle, as expected for a mixture of TpoR monomers and dimers. Corresponding changes in the peak width were observed, with a similar peak width for the monomers at (108 ± 1) photons and an increased

peak width for dimers at (175 ± 13) photons. From the peak integrals, a dimerization level of ~18% was estimated. TpoR monomers and dimers were confirmed by photobleaching events (Supplementary Movie 4 and 5, and Fig. 2g, h). Particles with low intensity typically disappeared upon bleaching, while 50% intensity remained in case of particles with high intensity. Very similar TpoR dimerization levels were obtained upon stimulation with its natural ligand thrombopoietin at saturating concentrations (Supplementary Fig. 6). In these experiments, JAK2 lacking the tyrosine kinase domain (JAK2ΔTK) was co-expressed to enable efficient TpoR dimerization without activating downstream signaling[41]. By classifying single-molecule localizations based on their intensities as monomers (< 625 photons) and as dimers (> 625 photons), we could detect an expected drop in the diffusion constant for dimers by approx. 25% (Supplementary Fig. 6i). The widths of the intensity peaks on live cells (Fig. 2d, f and Supplementary Fig. 6d, f) were in line with shot noise-limited simulations assuming an additional intensity variation based on axially modulated single-molecule trajectories with a standard deviation of 25 nm ascribed to the limited flatness of the basal plasma membrane (Supplementary Fig. 4e–g and 6j–l)[49]. These results highlight the capability of highly homogeneous TIRF illumination to robustly quantify monomer-

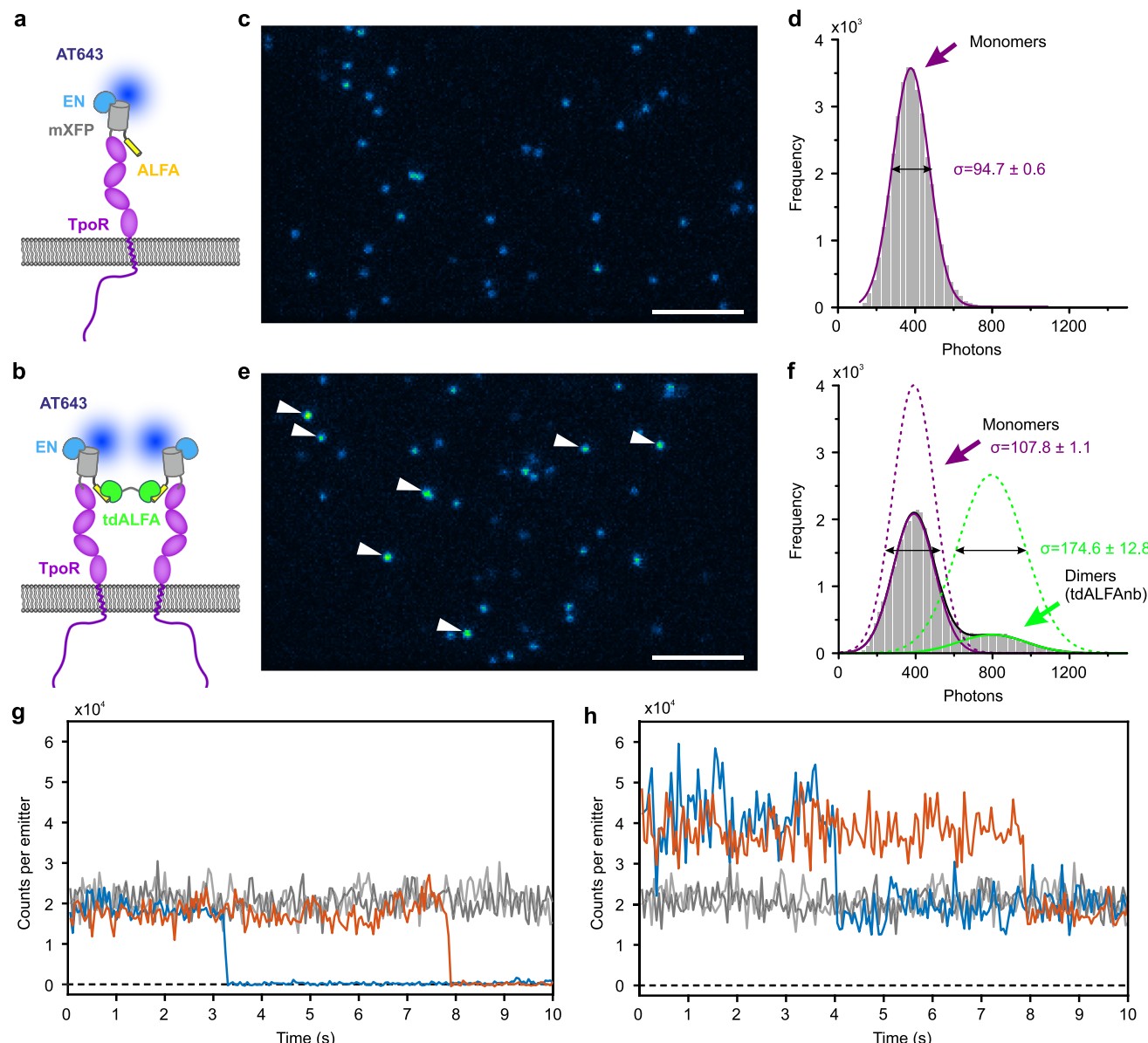

**Fig. 2 | Robust intensity analysis for quantifying receptor stoichiometries in live cells. a** Monomeric thrombopoietin receptor (TpoR) N-terminally fused to the ALFA-tag and a non-fluorescent monomeric EGFP (ALFA-mXFP-TpoR) was expressed in HeLa cells and cell surface-selectively labeled using anti-GFP nanobodies (EN) conjugated with ATTO 643. **b** Dimerization of ALFA-mXFP-TpoR via a tandem anti-ALFA nanobody (tdALFAnb). **c, d** Representative frame from time-lapse single-molecule imaging of monomeric ^AT643^EN-labeled ALFA-mXFP-TpoR in a HeLa cell **c** and intensity distribution analysis from 15 cells **d**. **e, f** Representative frame from time-lapse single-molecule imaging of ^AT643^EN-labeled ALFA-mXFP-TpoR in the presence of tdALFAnb **e** and intensity distribution analysis from 11 cells **f**. Scale bar in **c** and **e**: 5 μm. Dimers in **e** are highlighted by white arrows. Histograms were fitted with Gaussian or bimodal Gaussian, standard deviation σ is indicated. **g, h** Representative photobleaching events observed for ^AT643^EN-labeled ALFA-mXFP-TpoR in the absence **g** and presence **h** of tdALFAnb. Source data are provided as a Source Data file.

dimer equilibria in live cells by intensity analysis and to correlate receptor stoichiometries with their spatiotemporal dynamics.

## Quantitative smFRET analysis of receptor dimers in the plasma membrane

Single-molecule Förster resonance energy transfer (smFRET) offers even more potent opportunities for detecting and analyzing receptor dimerization in live cells. However, reliable detection and quantification of individual FRET signals diffusing in the plasma membrane remains highly challenging and only very few applications have been reported so far[37,41,42,50]. Homogeneous flat-top illumination by our TIRF-SIM setup could potentially enhance fidelity of quantitative smFRET detection. We again used ALFA-mXFP-TpoR as a model system, which

was labeled with donor and acceptor fluorophores by using a mixture of EN-labeled with Cy3B (^Cy3B^EN) and ATTO 643 (^AT643^EN) (Fig. 3a). We applied alternating laser excitation (ALEX) with 25 ms exposure for each laser for unambiguous smFRET detection by co-localizing donor and directly excited acceptor. Results from typical ALEX-FRET experiments acquired in the absence and in the presence of tdAL-FAnb are shown in Supplementary Movie 6 and Fig. 3b. Significant sensitized acceptor fluorescence signals were only detectable upon dimerizing of TpoR by the crosslinker. Quantitative analysis yielded dimerization levels of ~12 % (Fig. 3c, d), i.e., somewhat lower as compared to the dimerization levels estimated by single color intensity analysis. This discrepancy could be related to missing smFRET signals due to the low sensitized acceptor fluorescence levels. Receptor

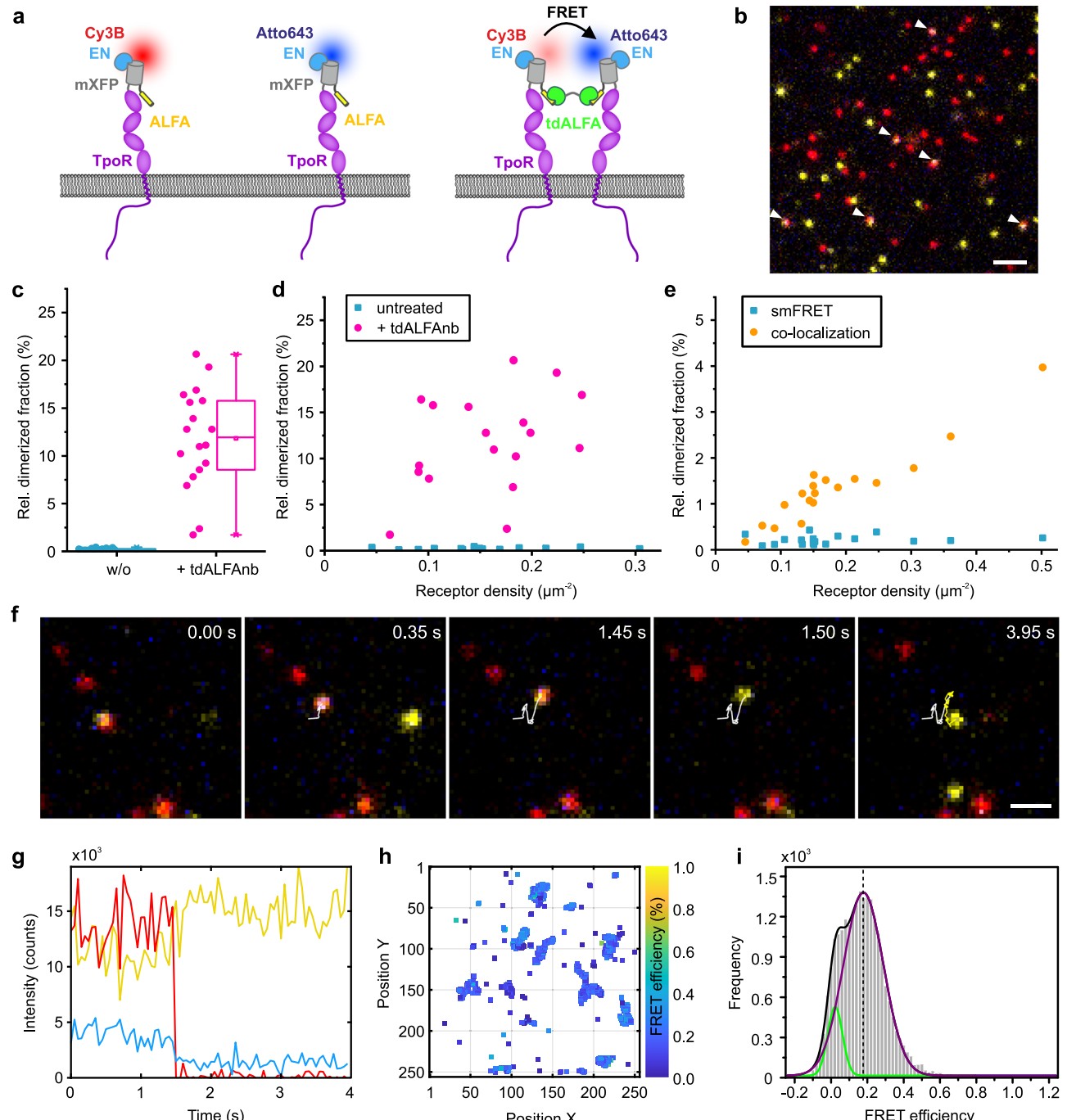

**Fig. 3 | Quantifying receptor dimerization by smFRET in live cells. a** ALFA-mXFP-TpoR expressed in HeLa cells was labeled by a mixture of $^{Cy3B}$EN and $^{AT643}$EN. Single-molecule ALEX-FRET imaging was performed in the absence and presence of crosslinker tdALFAnb. **b** Representative image from a time-lapse smFRET experiment of ALFA-mXFP-TpoR in the presence of tdALFAnb. Overlay of donor (yellow) and acceptor (red) channels (direct excitation) and sensitized acceptor channel (blue). FRET events are highlighted by white arrows. Scale bar: 2 μm. **c–e** Quantifying receptor dimerization by smFRET. Each data point represents the analysis from one cell with 18 cells measured for each condition. **c** Dimerized ALFA-mXFP-TpoR fraction in the absence and presence of tdALFAnb as quantified by smFRET. Box plot

indicates data distribution of the second and third quartiles (box), median (line), mean (square), and 1.5x interquartile range (whiskers). **d** Cell surface density-dependent analysis of same dataset shown in **c. e** Comparison of cell surface density-dependent analysis of false-positive ALFA-mXFP-TpoR dimerization in absence of tdALFAnb by single-molecule co-localization and smFRET. **f, g** ALEX-FRET tracking of a representative ALFA-mXFP-TpoR dimer **f** and changes in intensity upon acceptor photobleaching **g**. Yellow, donor; red, directly excited acceptor; blue, FRET signal. Scale bar in **f**: 1 μm. **h, i** FRET efficiencies of tracked receptor dimers **h** and FRET efficiency histogram obtained from 17 cells **i**. The histogram was fitted with a bimodal Gaussian. Source data are provided as a Source Data file.

density-dependent dimerization analysis based on smFRET confirmed a positive correlation for crosslinker-induced dimerization, while negligible dimerization levels in the absence of crosslinker were found, independent on the receptor density (Fig. 3d). By contrast, single-molecule co-localization analysis showed a substantial increase in

false-positive dimerization levels with increasing receptor density (Fig. 3e), highlighting more robust identification of molecular dimers by smFRET.

In addition to the detection of dimers, smFRET enables quantifying intermolecular distances of diffusing TpoR dimers in live cells by

analyzing FRET efficiencies. Spontaneous acceptor photobleaching during image acquisition confirmed donor recovery (Supplementary Movie 7, Fig. 3f, g), which, however, could not be systematically analyzed. We therefore determined smFRET efficiencies from the intensities of donor and sensitized acceptor emission using correction parameters calculated for the spectral detection conditions in our setup (Supplementary Fig. 7a–c). Correlation of FRET efficiencies with the apparent complex stoichiometry estimated from the intensities confirmed formation of TpoR dimers (Supplementary Fig. 7d). Based on these analyses, FRET efficiencies could be quantified for diffusing particles frame-by-frame, yielding spatiotemporal FRET efficiency maps as depicted in Fig. 3h and Supplementary Fig. 7e. FRET efficiency histograms could be fitted by a bimodal Gaussian distribution that discriminated residual signals without FRET (Fig. 3i). Indeed, localized molecules corresponding to these null FRET efficiencies were not part

of the FRET trajectories observed in spatial FRET efficiency maps (Fig. 3h). Thus, a mean FRET efficiency of 17.8% could be unambiguously determined, which corresponds to a donor-acceptor distance of 8.3 nm assuming the theoretical Förster radius of 6.4 nm. Such robust detection of FRET efficiencies below 20% highlight the huge potential of flat-top TIRF illumination for quantitative smFRET in live cells.

## Long-term single-molecule tracking and localization microscopy

In these single-molecule imaging experiments, we observed surprisingly high photostability upon flat-field illumination. We therefore explored more systematically long-term single-molecule tracking and localization microscopy (TALM) of TpoR in the plasma membrane, for which we have previously found free diffusion with a low immobile fraction[41]. For long-term tracking, TpoR fused to an N-terminal mXFP

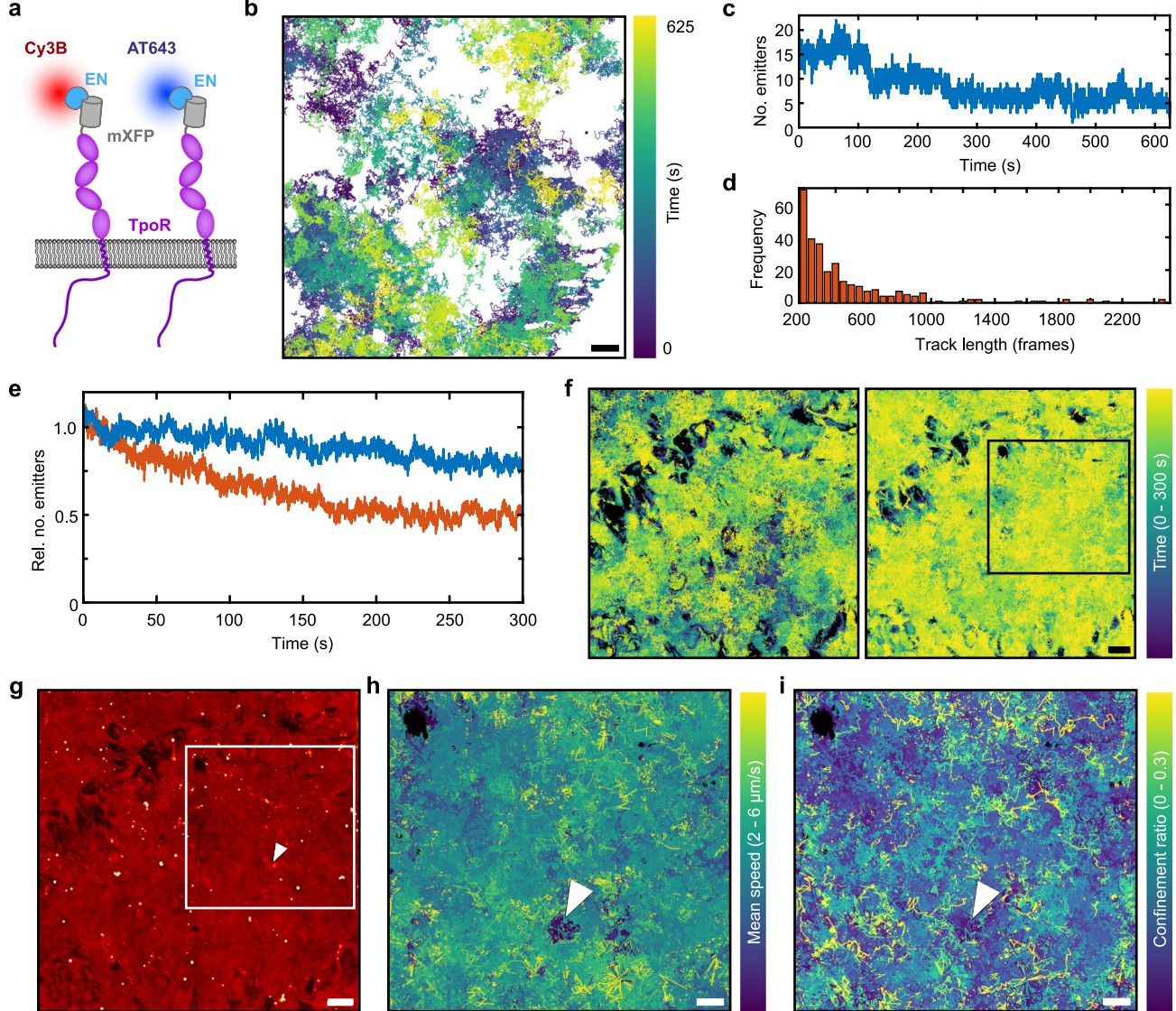

**Fig. 4 | Long-term dual color single-molecule localization microscopy. a** TpoR N-terminally fused to a non-fluorescent monomeric EGFP (mXFP) was expressed in HeLa cells and labeled with anti-GFP nanobodies conjugated with Cy3B and ATTO 643. **b** Typical single-molecule trajectories obtained from tracking for 625 s (25,000 frames). Scale bar: 2 μm. **c** Number of localized molecules per frame over the entire experiment (n = 2.59 × 10⁵ localizations). **d** Trajectory length histogram of trajectories with a length of ≥ 200 frames (5 s) (n = 285 trajectories). **e, f** Long-term dual-color single-molecule tracking and localization microscopy (TALM). **e** Normalized averaged number of localized molecules in the Cy3B (orange) and AT643 (blue)

channel versus time from three independent cells ($n_{Cy3B} = 1.01 \times 10^6$ and $n_{AT643} = 2.24 \times 10^6$ localizations). **f** Simultaneous TALM trajectories of TpoR labeled with Cy3B (left) and ATTO 643 (right) for 9,000 frames ($n_{Cy3B} = 16,502$ and $n_{AT643} = 38,538$ trajectories) from a representative experiment. Scale bar: 2 μm. **g** TALM image rendered from all localized molecules in the AT643 channel shown in **f. h, i** Trajectories color coded for mobility **h** and confinement **i** within the ROI highlighted in **g**. Scale bar: 1 μm. White arrow in **g–i** indicates area of reduced mobility and high confinement. Source data are provided as a Source Data file.

(mXFP-TpoR) was transiently expressed in HeLa cells and labeled by [AT643]EN as described above (Fig. 4a). To maximize tracking fidelity, cells with low cell surface expression (~0.2–0.3 molecules/μm²) were imaged. Highly robust detection and tracking of individual molecules was possible over the entire 25,000 frames (625 s) of the experiment (Supplementary Movie 8, Fig. 4b). With a typical signal-to-noise (S/N) ratio of 21 dB and a time resolution of 40 Hz applied in these experiments, the decay in the number of localized molecules during imaging was very slow, with ~30% of the initial number of molecules still being visible after 25,000 frames (Fig. 4c). This substantial enhancement of apparent photostability as compared to conventional 1-beam TIRF illumination can be explained by locally enhanced non-linear photobleaching due to over-illumination in some regions obtained by a Gaussian beam profile (Supplementary Fig. 8a–c). Robust single-molecule detection over extended time periods enabled long-term tracking of individual molecules (Fig. 4b). Even at the low cell surface density used in these experiments, tracking was limited by termination due to crossing of molecules rather than photobleaching. Typical trajectory length histograms showed a trajectory lifetime of ~10 s (400 frames), but a significant number of very long trajectories were obtained (Fig. 4d).

For systematically mapping receptor dynamics at the plasma membrane by TALM, cells expressing mXFP-TpoR at relatively high density were labeled with a mixture of [Cy3B]EN and [AT643]EN and imaged for 300 s by flat-top TIRF illumination at 561 nm and 640 nm (Supplementary Movie 9). Again, very long single-molecule imaging with sufficient S/N ratio was possible, with photobleaching of ~20% in the AT643 and ~50% in the Cy3B channel during 9000 frames of continuous acquisition at 30 Hz (Fig. 4e). Super-resolution TALM images rendered from localized TpoR yielded high-density coverage of the plasma membrane identifying relatively homogenous accessibility (Fig. 4f). For some regions, however, strongly reduced accessibility or even complete exclusion of TpoR was observed. These omitted regions correlated in both channels, corroborating that restricted accessibility was inherent to the biological specimen rather than an artifact caused by the stochastic interrogation by a limited number of probes. TALM localization maps highlighted these inaccessible areas, and furthermore identified "clusters" caused by molecules exhibiting highly restricted mobility for long time periods, which were frequently terminated by a brief and fast directed motion (Fig. 4g, Supplementary Movie 10). This behavior, which we interpreted to be related to TpoR endocytosis, could also be observed by average intensity projections across the entire image stack, showing characteristic comet-like structures (Supplementary Fig. 8e). Interestingly, no co-localization in both channels could be observed for these structures (Supplementary Fig. 8f), despite close to quantitative labeling of all cell surface TpoR at the experimental conditions (>70%[39]). These results indicate endocytosis of individual TpoR subunits rather than clusters as previously suggested for the interleukin-2 receptor[24], another member of the class I cytokine receptor family. The role of underlying cell morphology in receptor diffusion properties was further highlighted by the substantial differences in mobility and local confinement of TpoR that showed substantial spatial correlation (Fig. 4h, i). However, more, and high-resolution information on cellular structures are required for functional interpretation of these effects.

## Flexible combination of multi-color SIM and SMT imaging

To obtain such information, we implemented fast time-lapse SIM imaging and SIM reconstruction based on fairSIM and subsequent Hessian denoising[51–54]. This approach allows fast SIM acquisition at up to 200 Hz with high modulation contrast. By cycling between real-time single-molecule imaging at 30 Hz and short SIM acquisitions (Supplementary Fig. 12a), we achieved quasi-simultaneous SIM and SMT imaging in live cells (details in method section). Robust performance of

SIM deconvolution in all three channels and for the entire field of view was confirmed by imaging multi-color nanoparticles (100 nm diameter) randomly adsorbed at low density on a coverslip surface (Supplementary Figs. 9 and 10). Clear improvement in the resolution by a factor of two upon SIM deconvolution could be demonstrated by decorrelation analysis (Supplementary Fig. 9g-m) as well as by fitting experimental point spread functions (Supplementary Fig. 10b). Flat-field TIRF-SIM worked similarly well for the maximum field of view (approx. 40 × 40 μm²) without the additional 1.6x magnification, which is only required for single-molecule imaging at Nyquist-Shannon sampling (Supplementary Fig. 11). To explore the performance of time-lapse three-color SIM imaging in live cells, HeLa cells expressing TOM20-mEGFP were stained with Abberior Live Orange and SPY650-tubulin to stain the outer mitochondrial membrane, the crystae and the microtubule network, respectively. SIM imaging in GI mode yielded super-resolved images in all three channels with a time resolution of ~450 ms (~150 ms/channel), enabling to simultaneously monitor the dynamics of outer and inner membranes of mitochondria as well as their dynamic association to the microtubule network (Supplementary Movie 11, Supplementary Fig. 12). Likewise, combination of SIM and SMT was possible as exemplified by using TOM20-HaloTag labeled in two different colors as proof-of-concept experiments for correlating diffusion properties with nanoscale cellular structures (Supplementary Fig. 13, Supplementary Movie 12).

## Simultaneous SIM and SMT uncovers TpoR confinement by the cortical cytoskeleton

Having validated reliable SIM imaging, we attempted to correlate receptor diffusion in the plasma membrane with the dynamic structural organization of the cortical actin cytoskeleton. To this end, we co-expressed mXFP-TpoR with LifeAct fused to HaloTag (LifeAct-Halo-Tag) in HeLa cells. Upon labeling with HTL-JFX549 and [AT643]EN, single cells were imaged by SIM (JFX549 channel) and SMT (AT643 channel) in TIRF mode (Supplementary Fig. 7a). After SIM deconvolution, distinct features of the membrane-proximal actin cytoskeleton could be clearly discerned, while monitoring TpoR diffusion by SMT (Supplementary Movie 13 and Fig. 5a). Segmentation of the SIM images using machine learning (details in Materials and Methods)[55] yielded two types of cytoskeletal structures (Fig. 5b): a meshwork-type structure probably related to the membrane cytoskeleton (MSK) and a condensed structure probably related to focal adhesions. In line with this interpretation, the meshwork structure was highly dynamic while the condensed structure remained largely static (Supplementary Movie 13 and Fig. 5c). TALM images confirmed that TpoR was largely excluded from condensed actin structures, while homogeneous distribution in the meshwork structure was observed (Fig. 5d). In addition to mobile TpoR used for TALM rendering, a largely immobile fraction was observed as highlighted by a mean intensity projection stack obtained by averaging the 90 consecutive SMT frames between SIM acquisitions (Fig. 5e). These molecules were interpreted to be either in the process of endocytosis or already in endocytic vesicles.

Robust segmentation and classification of the cortical actin cytoskeletal structures based on machine learning enabled further analysis of the MSK meshwork dimension and dynamics. A broad distribution of corral geometry and size was observed, with a mean area per corral of 0.046 μm² which corresponds to a mean diameter of ~200 nm. (Fig. 5f, g). These observations are in line with cortical actin structures observed by super-resolution fluorescence microscopy in fixed cells[56,57] and by electron microscopy of unroofed HeLa cells[58]. Most of the meshwork actin structures dynamically changed from frame-to-frame (Supplementary Movie 13 and 14). In the zones with higher actin density close to focal adhesions that remained stable throughout the experiment, however, also the meshwork structure showed reduced dynamics. Diffusion properties of TpoR correlated with density and dynamics of the actin meshwork (Supplementary

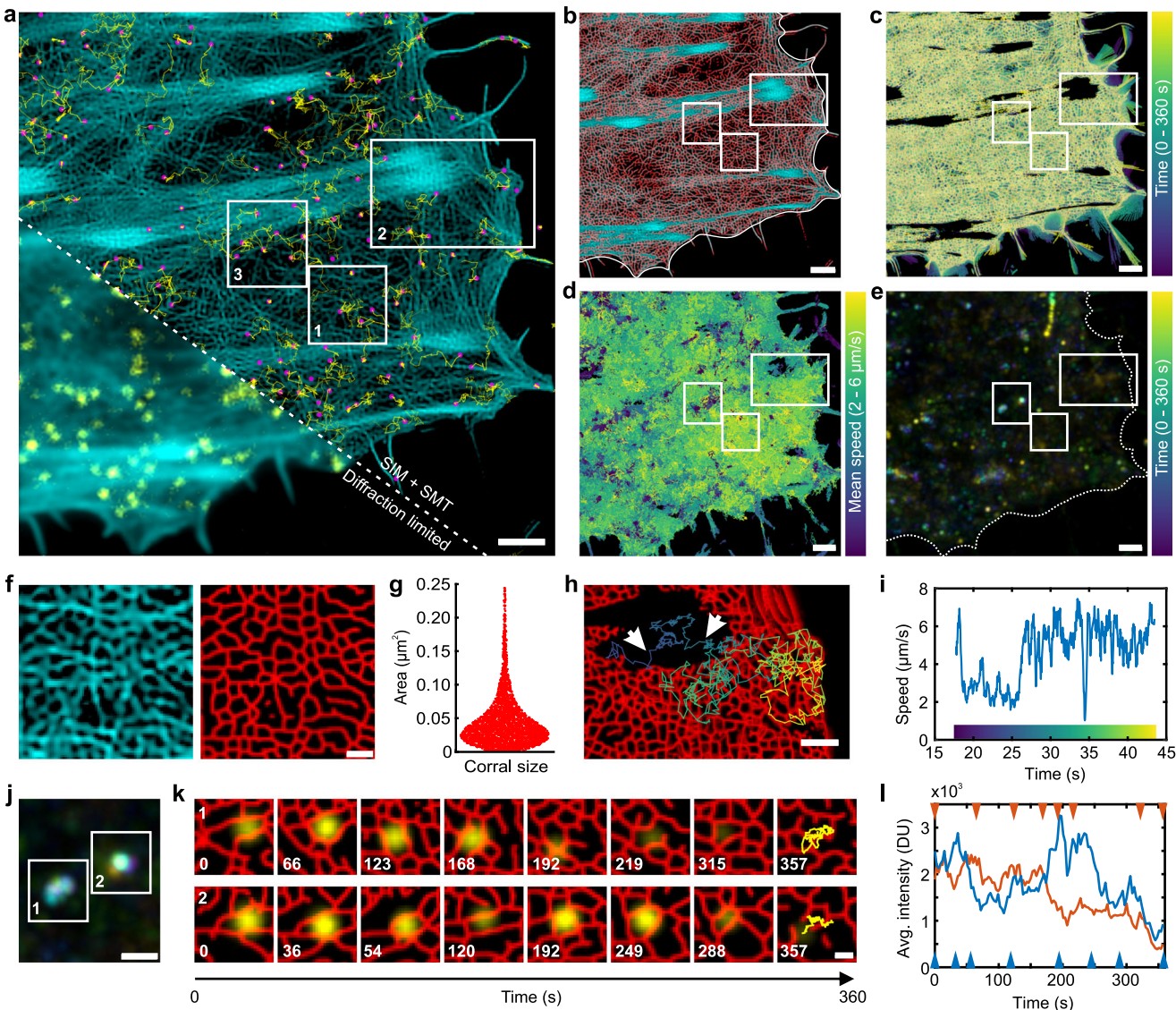

**Fig. 5 | Resolving receptor diffusion confinement within the cortical actin cytoskeleton by simultaneous TIRF-SIM and SMT. a** Single-molecule imaging of mXFP-TpoR diffusion (yellow) and the cortical actin cytoskeleton (cyan) of a representative cell. Magenta dots: single localizations, yellow lines: trajectories faded in time. A gamma correction (parameter = 0.5) as well as unsharp masking (radius = 1 pixel, mask weight = 0.6) was applied to actin channel. Scale bar: 2 μm. **b** Overlay of the original SIM image in cyan with the segmented actin meshwork in red. **c** Color-coded overlay of segmented meshwork structure from 120 consecutive SIM images. **d, e** Mobility of TpoR in the plasma membrane. **d** TpoR trajectories from 10,800 consecutive frames (360 s) color coded for mobility (n = 33,452 trajectories). **e** Color-coded time-lapse mean intensity projection from consecutive 90 single-molecule frames (120 time points; each 3 s). Scale bars in **b**−**e**: 2 μm. **f** Zoom into the cortical actin meshwork (ROI 1 in **a**): original SIM-deconvolved image (left) and segmented image (right). Scale bar: 1 μm. **g** Size distribution obtained for segmented meshwork (n = 3148). **h** Single-molecule trajectory covering different zones of actin structures (ROI 2 in **a**). Scale bar: 1 μm. **i** Differential mobilities within the trajectory shown in **h**. The color bar refers to the trajectory shown in **h**. **j**−**l** Two representative endocytosis events in the context of the cortical actin cytoskeleton. **j** Color-coded time-lapse mean intensity projection of TpoR in endosomal compartments (ROI 3 in **a**). **k** Overlay of segmented SIM images (red) and TpoR (yellow) of ROIs in **j**. The last images show the vesicle trajectory. Time stamp in seconds. Scale bar: 200 nm. **l** Changes in TpoR intensity during endosomal trafficking of top time series (orange) and bottom time series (blue) in **k**. Arrows indicate time stamps of images shown in **k**. Source data are provided as a Source Data file.

Movie 14, Fig. 5i). Even at the video-rate time resolution of SMT aplied in these experiments, confinement by the actin meshwork and hopping between individual corrals can be discerned.

### Tracking endocytosis of signaling complexes by simultaneous SIM and SMT

Simultaneous SIM and SMT furthermore enabled following endocytic events in the context of cytoskeletal nanostructures at single receptor level. Time-lapse mean intensity projections identified TpoR that remained immobilized over extended time periods before being removed from the cell surface (Fig. 5j, k, Supplementary Movie 15). For these receptors, residual mobility correlated with changes in the MSK meshwork, possibly being related to the active role of actin in pinching of endocytic vesicles. Accordingly, the TpoR signal disappeared gradually rather than showing stepwise photobleaching (Fig. 5l) which could be confirmed on single-particle level at full temporal resolution (Supplementary Fig. 14 and Supplementary Movie 16). These observations are in line with the high turnover rates of class I/II cytokine receptors at the plasma membrane even at resting state, which have been related to diverse mechanisms of endocytosis[21,59,60].

To confirm that long-term immobilization of TpoR was related to endocytosis, we co-expressed StayGold-2xFYVE (SG-2xFYVE) for

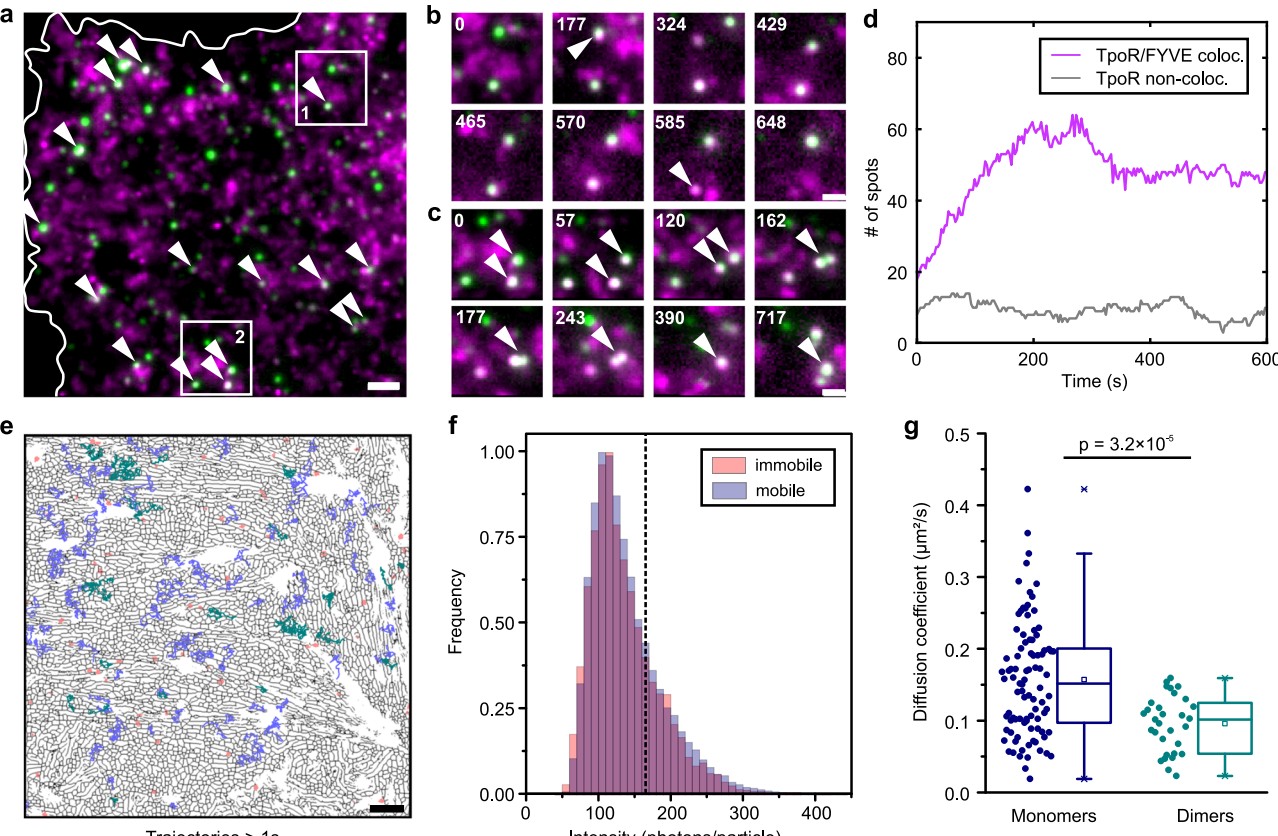

**Fig. 6 | Spatiotemporal dynamics of active TpoR signaling complexes.**
**a**–**d** Long-term single-molecule imaging of TpoR in HeLa cells co-expressing SG-2xFYVE for labeling of endosomes. **a** Overlay of a single image of SG-2xFYVE (green) with a mean intensity projection of TpoR (from 90 consecutive frames magenta). White arrows indicate co-localizing signals. Scale bar: 2 μm. **b**, **c** Time-lapse images for ROI 1 (**b**) and ROI 2 (**c**) highlighted in **a** (time stamp in seconds). **b** White arrows indicate two events of starting co-localization (upper endosome) and vanishing of signal (lower endosome). **c** Two endosomes fuse during time-lapse (white arrows). Scale bars: 1 μm. **d** Time-dependent changes in the co-localization of immobile TpoR with FYVE-positive endosomes compared with the non-co-localized fraction. **e** Simultaneous SIM imaging of LifeAct-SG and single-molecule imaging of TpoR in the presence of TPO. WEKA segmentation of the LifeAct-SG SIM image is shown in black (first time point only). Trajectories longer than 1 second are shown for first 12 seconds of the experiment color coded for low intensity (< 165 photons/particle, blue), high intensity (> 165 photons/particle, green) and immobile (orange). Scale bar: 2 μm. **f** Single-particle intensity histogram for mobile (blue) and immobile (red) TpoR. Dashed line at 165 photons indicates intensity threshold to classify signals for diffusion analysis shown in **g**. Signals with an intensity < 165 photons were classified as monomers, signals with an intensity > 165 photons as dimers. **g** Distribution of diffusion coefficients on single-trajectory level for filtered monomers (93 trajectories) and dimers (31 trajectories). Box plots indicate data distribution of the second and third quartiles (box), median (line), mean (square), and 1.5 × interquartile range (whiskers). Statistics for **g** were performed using two-sample Kolmogorov–Smirnov test. Source data are provided as a Source Data file.

labeling endosomal compartments. Indeed, distinct co-localization and co-diffusion of immobile TpoR and FYVE was observed (Fig. 6a–c, Supplementary Movie 17). Interestingly, localization of TpoR in endosomes strongly increased over time (Fig. 6d), in line with the typical minute time scale of endocytosis kinetics of cytokine receptors. By contrast, the fraction of TpoR not co-localized with FYVE did not systematically change over time, which could be explained by a constant steady-state fraction of TpoR in the process of endocytic uptake.

Having confirmed that the majority of immobile TpoR is localized in endosomes, we investigated in more detail the dynamics of active TpoR signaling complexes in the presence of full-length JAK2 and stimulated with TPO[41]. To this end, we performed long-term single-molecule imaging of nanobody-labeled TpoR after adding TPO at a concentration sufficient to saturate all high-affinity binding sites. By co-expression of LifeAct fused to StayGold (LifeAct-SG), simultaneous long-term SIM imaging of the cortical actin cytoskeleton was achieved (Fig. 6e, Supplementary Fig. 15a–f and Supplementary Movies 18–19). Overall changes in the diffusion properties of stimulated TpoR were modest, with the majority of receptors diffusing freely in the plasma membrane and being largely excluded from condensed actin structures (Fig. 6e, Supplementary Fig. 15d–e). Intensity analysis of all

mobile molecules, however, confirmed substantial dimerization, with only very low higher-order oligomerization (Fig. 6f). Supported by simulations (Supplementary Fig. 15g–h), we therefore analyzed the diffusion constants after sorting TpoR trajectories into populations with low (< 165 photons/particle) and high (> 165 photons/particle) intensity (Fig. 6g, Supplementary Fig. 15i and Supplementary Movie 20). A significant, ~40% decreased diffusion coefficient was observed for the high-intensity fraction, in line with previous observations on dimerized cytokine receptors[41]. The intensity histogram for immobile TpoR, indicated a minor shift to increased stoichiometries (Fig. 6f), highlighting that receptor clustering is not required for endocytosis. These examples highlight the exciting possibilities of simultaneous SIM and quantitative SMT imaging achieved with this specific setup.

## Discussion
Quantitative imaging the dynamic functional organization of proteins in the context of cellular nanostructures with sufficient spatiotemporal resolution remains a key challenge. To tackle this challenge, we here have introduced a SIM setup that deploys an SLM to flexibly achieve flat-field homogenous or structured illumination in up to three

spectral channels for combined single-molecule imaging and super-resolution SIM. Many different modalities of homogenous TIRF microscopy were described based on continuous TIR ring illumination in the back focal plane[61,62], galvometric laser scanning (spinning TIRF)[63–65] or multi-angular TIR illumination using digital mirror devices[66]. Combined with beam-shaping devices as used for standard TIRF microscopy[43,67], these techniques would generate very homogenous flat-field TIR illumination. A recently published hybrid scanning and widefield method called Adaptable Scanning for Tunable Excitation Region (ASTER) reaches shadowless flat-field TIR illumination even for tunable big field of views[68]. All these techniques could switch between HILO, GI or TIR illumination, and the spinning TIRF and the ASTER method are commercially available. Nonetheless, advanced applications such as real-time multi-color single-molecule tracking or ALEX-smFRET require precise high-speed control of evanescent field's penetration depth for multiple channels and advanced hardware synchronization of cameras, lasers and active optical or mechanical components. Apart from the SLM and an AOTF for laser blanking, we are using only passive optical elements which can be easily controlled by open-source software (e.g. micro-manager) and simple digital trigger signals. Using an SLM offers highest flexibility in choosing illumination mode and high-speed acquisition sequences. Since the first publication about video-rate TIRF-SIM microscopy[69], FLCOS-SLMs were an integral part of high-speed SIM systems. By introducing passive polarization rotation based on quarter wave plates and azimuthal pizza polarizers, these systems became easier to operate and were more cost-effective[51,52]. In addition, open-source user-friendly software for fast linear SIM deconvolution is available[53]. Although recent developments demonstrate, for example, scanner-based systems[70] or fiber optic configurations[71] to achieve fast TIRF-SIM, these systems are more complex to synchronize or less versatile, respectively, to achieve multimodal multi-color high-speed imaging. Recently, digital mirror devices (DMDs) were introduced as a promising alternative to SLMs for high-speed SIM applications[72–76], but robust multi-color applications with more than two colors are still challenging. Furthermore, new developments in image denoising will provide more opportunities to apply fast SIM in biological samples[54,77]. Exploiting a fast FLCOS-SLM in our setup, time-lapse SIM imaging in all three channels was possible with high time resolution up to 50 ms/channel. SIM was readily combined with homogeneous-field single-molecule imaging, enabling to simultaneously capture single-molecule diffusion properties and rapid changes of cellular nanostructures.

High homogeneity of TIRF illumination in conjunction with fast super-resolution SIM imaging opened exciting possibilities for live-cell imaging of plasma membrane dynamics. We demonstrate unambiguous discrimination of receptor monomers and dimers based on single-particle intensity analysis and by smFRET even with FRET efficiencies below 20%. Robust quantification of FRET efficiencies enabled discriminating between random co-localization and molecular dimers with high fidelity. Thus, receptor dimerization could be reliably detected even at elevated cell surface densities, at which a significant fraction of significant false-positive dimers was detected by co-localization. Strikingly, homogenous excitation strongly reduced photobleaching, thus enabling long-term single-molecule tracking of cell surface receptors and systematically mapping diffusion properties across the surface of the plasma membrane. We thus clearly identified zones that showed reduced mobility and accessibility, that we hypothesized to be caused by the cortical actin cytoskeleton. The critical role of this "membrane skeleton" (MSK) in defining receptor dynamics has been first put forward by Edidin & Kusumi[78,79] and since then validated and refined by numerous studies[2,80,81]. The potential of TIRF-SIM to visualize cytoskeletal nanostructures in the plasma membrane of live cells has been demonstrated[82,83], but combination of fast SIM and SMT to uncover receptor dynamics in the context of nanoscale MSK confinement has not yet been described. By fast TIRF-SIM

imaging in conjunction with segmentation based on machine learning, we were able to resolve the MSK meshwork down to corral sizes of ~100 nm diameter. The sizes and geometries are overall in agreement with what has been observed with SMLM in fixed cells[56,57] or by electron microscopy[58]. While not reaching the resolution of these techniques, our approach is capable to capture the nanoscale dynamics of the MSK. In conjunction with long-term TALM, we therefore could correlate diffusion properties of receptors with the dynamics of the MSK. While receptor diffusion is clearly inhibited by less dynamic actin structures, we find that the motion of endocytic vesicles at the plasma membrane depends on MSK reorganization. Long-term single-molecule imaging enabled reliable identification and tracking of individual endocytosed receptors and signaling complexes. While focusing here on receptor dynamics in the plasma membrane, we also demonstrate the capabilities for super-resolved imaging of organellar and sub-organellar dynamics in combination with SMT. Taken together, our microscope offers powerful and versatile capabilities for quantitative imaging dynamic cellular processes at nanoscale and single-molecule resolution.

## Methods

### Materials
Anti-GFP nanobodies "enhancer" labeled with Cy3B or ATTO 643 maleimide, respectively, via an engineered cysteine residue at the C-terminus were produced as described previously[39]. For artificial dimerization of ALFA-mXFP-TpoR, we designed a tandem anti-ALFA nanobody (tdALFAnb) by cloning two anti-ALFA nanobody repeats separated by a linker of 7 amino acids (ESFSGGS) and a C-terminal hexahistidine (His6)-tag into pET-21a. tdALFAnb was expressed in *E. coli* SHuffle T7 and purified to homogeneity by immobilized metal affinity chromatography and size exclusion chromatography using standard protocols.

### Sample preparation for in vitro reHaloTag imaging
Glass coverslips were silanized with (3-glycidyloxypropyl)trimethoxysilane (GOPTS), coated with diamino-PEG 2000 and functionalized with tris-(nitrilotriacetic acid) tris-NTA, as described before[47]. After loading of Ni(II) ions, 4 µM reHaloTagF fused to a hexahistidine (His6)-tag[46], was incubated for 30 min at room temperature. After washing, HTL-MaP555 diluted to 50–200 pM in PBS was added and imaged.

### Cell culture of HeLa cells
HeLa cells were cultivated at 37 °C and 5% $CO_2$ in MEM's Earle's medium with stable glutamine supplemented with 10% fetal bovine serum (FBS), non-essential amino acids and HEPES buffer without addition of antibiotics.

### Sample preparation for live cell imaging
HeLa cells were trypsinated and transferred to a 60 mm cell culture dish one day before transfection. One day prior to microscopy experiments, HeLa cells were transiently transfected via calcium-phosphate precipitation[84] for 6 h before washing with PBS, in order to transiently express recombinant plasmid DNA (Table 1). After transfection, the cells were trypsinated and transferred onto 24 mm glass coverslips coated with a poly-L-lysine-graft (polyethylene glycol) copolymer functionalized with RGD using a concentration of 1 mg/ml in PBS to support adhesion and minimize non-specific binding of fluorescent nanobodies[85]. For microscopy experiments, glass coverslips were mounted into custom-designed microscopy chambers containing 1 mL of phenol-red free medium. ALFA-mXFP-TpoR and mXFP-TpoR receptors were labeled by adding 3 nM of anti-GFP nanobodies (EN), conjugated with Cy3B or ATTO 643, respectively. HaloTag constructs were labeled with Janelia Fluor® dyes JFX549 or JFX646, respectively (Table 2). To achieve detection of intensities for smFRET efficiency and single-color stoichiometry analyses with

## Table 1 | Recombinant DNA

| Denomination | Plasmid name | Source |
|---|---|---|
| tdALFAnb | pET-21a tdALFAnb-H6 | this study (Addgene #222944) |
| mXFP-TpoR | pSems leader mXFP-TpoR | Wilmes et al.[41] |
| ALFA-mXFP-TpoR | pSems leader ALFA-mXFP-TpoR | this study (Addgene #222945) |
| Jak2-tdmCherry | pSems JAK2-tdmCherry | this study (Addgene #222946) |
| Jak2ΔTK-mEGFP | pSems JAK2 (1-827)-mEGFP | Wilmes et al.[41] |
| LifeAct-HaloTag | pSems LifeAct-HaloTag | Wilmes et al.[101] |
| LifeAct-StayGold (LifeAct-SG) | pSems LifeAct-StayGold<br>• StayGold from Addgene #185823[102] | this study (Addgene #222947) |
| StayGold-2xFYVE (SG-2xFYVE) | pSems StayGold tdmFYVE<br>• StayGold from Addgene #185823[102]<br>• 2xFYVE from Addgene #140047[103] | this study (Addgene #222948) |
| TOM20-mEGFP | pSems TOM20-meGFP | Appelhans et al.[104] |
| TOM20-HaloTag | pSems TOM20-HaloTag | Appelhans et al.[104] |
| farnesyl-GFP | pSems farnesyl-GFP | this study (Addgene #222949) |

## Table 2 | Nanobodies, proteins and dyes

| Protein (denomination)/Dye | Source |
|---|---|
| Anti-GFP nanobody enhancer (EN) | Sotolongo Béllon et al.[39] |
| tandem anti-ALFA nanobody (tdALFAnb) | this study |
| reHaloTagF-linker-aGFPnb-enhancer-H6 (reHaloTagF) | Holtmannspötter et al.[46] |
| Gattaquant PAINT 40RY | Gattaquant |
| Cy3B Maleimide | GE Healthcare |
| ATTO 643 Maleimide | ATTO-TEC |
| ATTO 655 | ATTO-TEC |
| HTL-MaP555 | Gift by Kai Johnsson |
| Fluorescein | Sigma Aldrich |
| Texas Red | Sigma Aldrich |
| Abberior Live Orange | Abberior |
| SPY650-tubulin | Spirochrome |
| HTL-JFX549 | Janelia Materials |
| HTL-JFX646 | Janelia Materials |

minimal bias, the nanobodies were washed out after 5 min incubation and new medium was added for the experiment. For stoichiometry analysis, smFRET and long-term tracking experiments the medium was supplemented with an oxygen scavenger and a redox-active photoprotectant to minimize photobleaching[86]. All live cell imaging was performed at 25 °C.

### Sample preparation for channel registration
High resolution channel registration was achieved by imaging 100 nm TetraSpec™ microspheres (T7279, ThermoFisher) immobilized on a 24 mm coverslip. To this end, single coverslips were cleaned by plasma for 10 min (Femto A, Diener) and coated with poly-L-lysine (PLL) using a concentration of 1 mg/ml in PBS. After an incubation time of 10 min, coverslips were rinsed with MilliQ water and dried by pressurized air. TetraSpec™ microspheres were diluted 1:50 in MilliQ water and 20 μl were applied to a single coverslip to let the solution completely dry. Finally, we applied liquid embedding medium from the PS-Speck™ microscope point source kit (P7220, ThermoFisher) and made a sandwich with a second coverslip.

### Flat-field structured illumination microscope
The microscope design is based on high-speed two-beam structured illumination microscopy (SIM) combined with an overall flat-top beam profile (Supplementary Fig. 1a)[43,51,52]. Three lasers, a 488 nm diode laser (Sapphire 488-400 CW CDRH, Coherent), 561 nm fiber laser (2RU-VFL-P-2000-560-B1R, 2000W, MPB communications) and a 642 nm fiber laser (2RU-VFL-P-2000-642-B1R, 2000W, MPB communications) were coupled into a single-mode polarization maintaining fiber (0.7-FCP8-P0, Qioptiq) to generate a TEM00 Gaussian beam profile. An AOTF (AA.AOTFnC-400.650-TN, AA Opto Electronic) was used for laser power attenuation and fast laser line switching and blanking. Each laser line was expanded before entering the AOTF to stay below the damage threshold of the AOTF. Due to different beam diameters of laser lines, we used different sets of lenses. The 488 nm laser was expanded by a factor of 3.12 (L1 and L2; C240TMD-A and AC127-025-A, Thorlabs), the 560 nm by a factor of 2.27 (L3 and L4; C220TMD-A and AC127-025-A, Thorlabs). Same lenses were used for the 642 nm laser (L5 and L6). After the AOTF, combined beams were demagnified by a factor of 3.2x (L7 and L8; #47-666 and # 47-660, Edmund Optics) to obtain correct input beam diameter for fiber coupling. The beam at the fiber output was collimated by an achromatic 50 mm lens (L9; AC254-050-A, Thorlabs). A πShaper (AdlOptica, #12-644, Edmund Optics) converted the Gaussian beam profile into a flat-top profile. Pure linear polarization was achieved by a Glan-Taylor calcite polarizer (P; GT10-A, Thorlabs). The 6 mm flat-top beam was expanded with two lenses (L10 and L11; #49-357 and #67-650, Edmund Optics) by a factor of 2.08 to illuminate the chip of a binary spatial light modulator (FLCoS-SLM, QXGA, 2048 x 1536 pixels, pixel size 8.2 μm, Forth Dimension Displays). An adjustable iris (I; CP20D, Thorlabs) is used to clip the beam size to 12 mm to underfill the SLM chip (shorter edge length 12.6 mm). By loading binary line patterns to the SLM, we introduced phase diffraction gratings as typically used for classical SIM. We used three equidistant orientations (0°, 120°, 240°) and three phases (0, $1/3\pi$, $2/3\pi$) throughout all experiments. The SLM is conjugated to the focal plane of the 100× oil immersion objective (UPLAPO100xOHR, Olympus) with a demagnification factor of 0.0045× leading to a pixel size of 37 nm in the focal plane. The Fourier-plane of the first 250 mm lens (L12; #47-647, Edmund Optics) after the SLM was used to block the DC component and higher diffraction orders by a simple home-built aluminum mask (M, holes punched in black aluminum foil) mounted on a rotating mount. Azimuthal polarization was guaranteed by a quarter wave plate (QWP, AQWP10M-580, Thorlabs) in combination with a segmented polarizer ("pizza polarizer") having 12 segments (PP; colorPol VIS 500 BC3 CW01, Codixx). First-order diffractions are imaged to the back focal plane of the objective by two relay lenses (L13; #47-647, Edmund optics, and L14; ACT508-400-A, Thorlabs). We used two identical 3 mm polychroic mirrors (DM3 and DM4; zt405/488/561/640rpc, Chroma, 3 mm) to minimize wave front distortions and to guarantee azimuthal polarization state after reflection. The first dichroic mirror (DM3) is positioned such that the s- and p-axes are switched compared to the

second dichroic mirror (DM4) located in the filter turret of the microscope[51,87]. The back focal plane could also be imaged by an inspection CMOS camera (BFP Cam; Alvium 1800 U-319m, Allied Vision) by using a flip mirror (FM) and two relay lenses (L15 and L16; AC254-200-A and LA1229-A, Thorlabs). We used a commercial motorized microscope body (IX-83, Olympus) equipped with full live cell imaging periphery (cellVivo incubator, Olympus) to control temperature, humidity and CO$_2$. The sample is mounted on a motorized microscope stage (SCAN IM 120 × 80, Märzhäuser). A hardware autofocus system (IX3-ZDC2, Olympus) allowed for long-term single-molecule imaging without axial drifts. Please, note that the hardware autofocus system was not compatible with SIM because the additional dichroic mirror of the IX3-zdc2 between objective lens and main dichroic mirror deteriorates polarization state and modulation depth of structured illumination. Fluorescence emission was filtered by a rejection band filter (F1; zet405/488/561/647 TIRF, Chroma) and imaged by an additionally 1.6x magnification (IX3-CAS, Olympus) on two cameras mounted by a dual camera adapter (TwinCam, CAIRN Research). Single-molecule data was acquired by an EMCCD camera (iXon Ultra, Andor), SIM data by a fast sCMOS camera (ORCA Fusion-BT, Hamamatsu). Unless otherwise stated, single-molecule imaging was performed with an EMCCD preGAIN of 3 and an EM GAIN of 300. A silver mirror (F21-005, 2 mm, AHF analysentechnik) or appropriate dichroic mirrors (DM5; H 560 LPXR superflat or H643 LPXR superflat, 2 mm, AHF analysentechnik) can be added to the TwinCam for SIM acquisitions only or for combined single-molecule and SIM imaging. Final pixel size was calibrated to 101.5 nm for the EMCCD camera and to 75.8 nm using 2 × 2 pixel binning on the sCMOS camera. If a bigger field of view for SIM was required, we removed the additional 1.6x magnification and imaged without pixel binning to achieve a calibrated pixel size of 64.1 nm. Each camera is equipped with an image splitter (QuadView, Photometrics) for fast multi-color imaging. Identical QuadView filter sets (FS) with three dichroic mirrors (DM6, 480dcxr; DM7, 565dcxr; DM8, 640dcxr; all Chroma) and single bandpass filters for the channels green (F3; BrightLine HC 520/35, Semrock), orange (F4, BrightLine HC 600/37, Semrock) and red (F5, BrightLine HC 685/40, Semrock) were applied.

## Software environment and hardware synchronization

Microscope control, image acquisition and hardware synchronization were realized with the open-source software micro-manager (www.micro-manager.org)[88] (Supplementary Fig. 1b). The SLM board was controlled and programmed in MetroCon V4.1 (Forth dimension displays). Phase gratings were loaded as binary bitmap files to the SLM board. Each image file is linked to a sequence file via a so-called running order (RO) (Supplementary Fig. 1c). ROs define the sequence of displayed images on the SLM which can be triggered (e.g. by camera) or send trigger signals to external devices. The sequence file determines timing of image display and illumination control (LED enable). To prevent SLM damage, each image needs to be inverted and displayed again for the same amount of time (DC balancing). Since an inverted pattern generates the same diffraction pattern, both (original and inverted) displayed images were illuminated (LED enable high). Image display switching time was 310 μs. Fastest available sequence file allowed for 500 μs image display leading to a shortest cycle time of 1.62 ms. We used sequence files with 500 μs, 1 ms, 2 ms, and 5 ms image display timings for different camera exposure times. Display of SLM gratings was synchronized with laser blanking and camera triggering by using simple programmable digital micro-controllers (Arduino UNO) (Supplementary Fig. 1b). In case of single-molecule imaging, the EMCCD acted as the master, running in streaming mode and triggering the SLM via its exposure high signal (Cam Fire). The SLM in turn triggered an appropriate laser line via the "LED enable" signal using the Arduino and hardware-based sequencing in micro-manager. Maximum acquisition rate at full frame (512 × 512 pixels) was 50 Hz. Two-color

imaging (561 & 642 nm laser) allowed for cropped acquisition (512 × 256 pixel) at maximum 100 Hz. SIM acquisition with the sCMOS camera required external triggering to synchronize SLM grating display with global exposure timing of the camera's rolling shutter. Here, the SLM was the master triggering camera (global reset edge mode) as well as the appropriate laser line via an Arduino. Field of view corresponded to 1200 × 1200 pixels (600 × 600 pixels at binning 2 × 2) requiring a readout time of 6 ms. Single-color or dual-color imaging (561 & 488 nm laser) halved the readout time by 3 ms and set the maximum frame rate to 216 Hz ( = 1/(1.62 ms + 3 ms)). Calculation of SLM patterns is based on a FIJI[89] plugin (https://github.com/fairSIM/fast SIM-GratingSearch). The algorithm is searching for the best combinations of gratings for each laser wavelength with lowest degradation by unwanted diffraction orders. The code is based on Matlab code published by Lu-Walther et al.[51]. The period of the illumination line pattern dictates the resolution gain factor (RGF) which can be maximum 2 for linear SIM. Furthermore, the period defines the incident angle of the two-beam interference in the focal plane. Since we are imaging live cells (refractive index $n = 1.36$) in aqueous medium ($n = 1.33$) on glass ($n = 1.518$) with an oil immersion objective, one can switch between total internal reflection (RGF = 1.900–1.950), grazing incidence[45] (RGF = 1.875–1.900) and highly inclined and laminated optical sheet (HILO)[44] illumination (RGF < 1.875). Homogenous illumination was achieved by cycling all 9-line patterns during a single exposure time. Single-molecule imaging and SIM at the plasma membrane was achieved with phase gratings based on a gain factor RGF = 1.95 (TIRF-SIM), super-resolution SIM inside cells on a gain factor RGF = 1.875 (GI-SIM). For quasi-simultaneous single-molecule imaging and SIM, we programmed (SLM) a loop of real-time single-molecule imaging at 30 frames per second for a couple of seconds followed by a single fast SIM acquisition (9 images). Here, the EMCCD was the master clock triggering the SLM. During single SIM acquisitions, the SLM took over the triggering of the sCMOS. After a full SIM set, the SLM waited for another trigger of the EMCCD camera and thus guaranteed a precisely defined timing.

## Microscope Alignment

**Excitation path.** All three laser lines were coupled into a single-model polarization maintaining fiber as described before. We only used the 560 nm laser for the following alignment and checked chromatic aberrations afterwards. The SLM was running a blank image to work as a simple mirror. The dim diffraction pattern by the pixelated display did not interfere with alignment. First task was the collimation of the laser beam coming out of the fiber and alignment to the main optical axes without any lenses defined by the first mirror, the SLM and the dichroic mirror DM3 (Supplementary Fig. 1a). These mirrors were mounted on 4-axis (tip, tilt and xy-translation) kinematic mounts. Collimation of the beam was determined by a shearing interferometry (SI100, Thorlabs). It is important to keep the incident angle onto the SLM as small as possible (here 6°). Detailed alignment of a similar system was well described by Young et al.[87]. We marked the laser spot at the ceiling of the laboratory as a reference. Next, we aligned the πShaper using a special tip/tilt and two translational axes lens mount (4-axis πShaper mount M27, Edmund optics) in combination with alignment tools by the manufacturer (πShaper aligner, Edmund optics). Perfect alignment was confirmed by an inspection camera following instruction by the manufacturer (Supplementary Fig. 2a). Afterwards, we again checked alignment of the laser spot to the reference point and collimation by the shear interferometer. Next, we placed the first beam expander (L10 & L11) and set the polarizer in front of the expander to vertical polarization and maximum transmission confirmed by a laser power meter. A 30 mm cage system (Thorlabs) between L13 and DM3 and iris I2 was used to define the optical axis for reviewing alignment. Then, we placed the objective lens and L14 into the light path. Collimation was set by axial displacement of L14, xy-

translation of L14 moved the laser spot back to the reference point. Same procedure was used for L12 and L13. The next task was the alignment of the pizza polarizer (PP) without the mask (M) and quarter wave plate (QWP). Therefore, we used the back focal plane camera (BFP Cam) and the flip mirror FM and loaded a fast sequence of 9 SIM diffraction patterns onto the SLM. We rotated and translated PP so that each laser spot of the +/- first diffraction order is centrally hitting a pizza segment. Then, we placed the mask and aligned it to the laser spots. By cycling through all laser lines and between HILO and TIR mode we guaranteed that the holes in the mask are just big enough to not clip the laser beams. Then, we added the QWP and aligned it by rotation to obtain similar laser power for each of the three SLM pattern orientations used for SIM. Final alignment of the system was determined by using dye solution (Fig. 1 and Supplementary Movie 1) in combination with mirror DM3 to obtain flat-top homogenous signal on the camera as well as a symmetric output of the lasers inside the dye solution by visual inspection. Dye solution was containing $1-2\,\mu M$ concentration of fluorescein, Texas Red and ATTO 655 in PBS. Dye solution was added on a 10 min plasma-cleaned (Femto A, Diener) coverslip within a custom-designed microscopy chamber.

**Emission path.** Pre-alignment of the dual camera adapter, the two image splitters and the cameras were performed by using an alignment laser (CPS532-C2, Thorlabs) mounted directly onto the objective turret. A detailed description of this procedure can be found elsewhere[90]. After alignment of the excitation path, one need to verify conjugation of the SLM with the focus plane and the camera planes. Therefore, we are using again dye solution and iris I1. Since I1 is generating a sharp edge on the SLM, this edge was used to conjugate both cameras using a xyz-translational mount. Final alignment of both cameras was achieved by imaging 100 nm TetraSpec™ microspheres (ThermoFisher).

### DNA-PAINT imaging of DNA origamis

Commercial microscope slides with immobilized dual-color nanorulers (PAINT 40RY, GATTAquant DNA Nanotechnologies) ready for DNA-PAINT imaging were used to demonstrate super-resolution capability of our system. These nanorulers contain three docking sides with a distance of 40 nm each presenting a docking strand for a Cy3B-imager and an ATTO 655-imager strand. We recorded 30,000 frames for each channel with an exposure time of 200 ms using TIR illumination. Images were recorded in alternate excitation mode to prevent bleed through of Cy3B signals into the ATTO 655 channel. Focus drift was compensated by the Olympus hardware autofocus system. Experiment was performed at 25 °C. Laser power of the 561 and 640 nm laser was set to approx. 10 mW at the back focal plane of the objective resulting in approx. 500 W/cm² in the focal plane. EM gain of the camera was set to 50.

### PAINT of reHaloTagF coated surfaces

For PAINTing of reHaloTagF monolayers on functionalized glass surfaces, 4000 frames were acquired using an exposure time between 25–50 ms to ensure a sufficiently high density of single-molecule localizations. All experiments were conducted at 25 °C and focus was stabilized during the experiment by the Olympus z-drift compensation device. Illumination intensity was adjusted to obtain similar single-molecule intensities as in live-cell single-molecule tracking experiment (100–200 photons per particle and frame). Single-molecule intensity analysis was compared under different illumination and detection conditions with (Fig. 1d–g) and without image splitter optics (Supplementary Fig. 4). In case of full field of view of the EMCCD camera (512 × 512 pixel), we performed PAINT experiments with and without πShaper using just a single angle of structured illumination or a sequence of three angles. By replacing the spatial light modulator with a dielectric mirror, we generated simple single-beam TIRF illumination as typically provided in standard commercial TIRF systems. In most

cases, we observed two stable intensity populations for MaP555-HTL on these surfaces (Supplementary Fig. 4 and Supplementary Table 1).

### Simulations of PAINT and single-molecule tracking experiments

Single-molecule experiments were simulated with a single-particle tracking (SPT) simulation software SPT_simulator developed in-house using Matlab (R2020b, Mathworks) (Supplementary Fig. 3a). Imaging and emitter parameters are described in Supplementary Table 2. The software simulates single particles (monomers and fractions of oligomers with specified particle density) which are randomly distributed on the user-defined imaging area and can diffuse over time based on random walks. Free diffusion in each dimension is modeled by normally distributed random numbers multiplied with mean displacements $\bar{d}$ based on given diffusion constants $D$ and simulation frame interval time $\Delta t$:

$$\bar{d} = \sqrt{2D\Delta t}. \tag{1}$$

The user can define different diffusion constants for the monomeric and oligomeric particles. In case of simulating PAINT experiments, the diffusion constant was set to zero. Each emitter can stochastically bleach/unbind over time based on an averaged lifetime $\tau$ modeled by a single exponential decay. In case of a Gaussian illumination profile, single emitter intensities are adjusted in dependence to their lateral position with an overall mean intensity set to the predefined emitter intensity $S_e$. Furthermore, we modelled the axial excitation intensity profile by a single exponential decay with a user-defined penetration depth $d_p$ (Supplementary Fig. 3e). Each simulated particle gets an axial $z$ position and the predefined $S_e$ is corrected by:

$$S_e^{corr} = S_e e^{z/d_p}. \tag{2}$$

This correction guarantees that a user-defined single-emitter intensity is obtained for a specified axial position given by the mean axial position $z_0$ and its standard deviation. The software allows for an additional emitter intensity variation based on the axial emitter position within the evanescent field. This is useful to model spatial variations in live-cell single-molecule tracking experiments representing e.g. plasma membrane ripples. The variation in axial positions along each trajectory is modelled by a normally distributed random number generator defined by the mean axial position $z_0$ and the standard deviation $\sigma_z$. Please, note that we didn't model the membrane, but we only introduced an additional variation on single trajectory level to account for signal intensity variations due to the steep axial decay of the evanescent field's excitation intensity (Supplementary Fig. 3f). To obtain smooth trajectories, we defined equidistant nodes (distance of nodes is set to $2/\bar{d}$) for random axial positions and interpolated remaining z positions by a spline function. Then, single-emitter fluorescence generation was modeled by a Poisson process based on the corrected mean signal intensity $S_e^{corr}$ with respect to the illumination profile and z position as described before. In order to simulate imaging frames, each programmed random trajectory is transferred to a 5 nm fine two-dimensional pixel grid (super-sampled version of final image frame) and each emitter is modeled by a Gaussian point spread function (PSF) specified by the given signal intensity $S_e^{corr}$ and a standard deviation derived from the Abbe limit $d_{Abbe}$:

$$\sigma_{PSF} = 1.3 \frac{d_{Abbe}}{2.355} = 0.276 \frac{\lambda_{em}}{NA}. \tag{3}$$

Here, $\lambda_{em}$ is the emission wavelength of the dye and NA the numerical aperture of the objective. The prefactor (1.3) was experimentally determined and corrects for effective numerical aperture and resolution. After placing of each emitter, each super-sampled image frame is down-sampled to the final pixel size and the mean background

intensity is added to each pixel value. Afterwards, each pixel value in units of photons is transferred to photoelectrons by the detector's quantum efficiency ($QE$) and pixel noise was modeled by a Poisson process with an additional noise factor of $\sqrt{2}$ to account for the excess noise of the EMCCD camera. Finally, photoelectrons are converted to digital counts by the camera specific electron conversion factor (ecf) and additional normally distributed noise is added based on the camera offset and noise level. The software generates a user-defined number of tif-stacks and ground-truth tables as well as corresponding Matlab m-files containing ground-truth data and all parameters. Parameters used for simulations in this study are listed in Supplementary Table 3.

### Single-molecule analyses

**Localization and tracking.** Single-molecule localization and tracking for stoichiometry analysis and smFRET based dimerization analysis was carried out using self-written Matlab (R2013a, MathWorks) code (SLIMfast)[41]. Channel registration based on affine transformation is performed within SLIMfast by images of 100 nm TetraSpec™ microspheres (ThermoFisher). Single molecules were localized using the multiple-target tracing algorithm[91] without deflation loops and PSF radii of 1.2 pixel or 1.3 pixel, for 561 nm excitation or 640 nm excitation, respectively. Tracking is based on utrack[92] and was done using default parameters. Based on the simulations in this study, we found that SLIMfast slightly underestimates single-molecule intensities at higher signal levels. At 100 photons per single emitter, the error is less than 1%, at 400 photons underestimation is approx. 5% (cp. Supplementary Fig. 3g and 6j–l). Nonetheless, this systematic error did not affect the conclusions of our findings.

**Intensity analysis and performance benchmarking using reHaloTag PAINTing.** Single-molecule localization was performed in SLIMfast as described above. In case of the smaller field of view using the image splitter (256 × 256 pixel, Fig. 1d), tracking could be accomplished directly in SLIMfast. Full field of view datasets (Supplementary Fig. 4) were further analyzed using TrackMate 7.10.2. in FIJI[89,93,94] (cp. section "Long-term localization and tracking" below). We used stringent parameters for tracking immobile signals: tracking search radius was set to 100 nm and gap closure was not allowed. Trajectories were filtered by trajectory length ≥ 3 frames and exported to Matlab for further analysis. For each trajectory, we removed the first and the last particle because these signals did not coincide with full exposure time of the camera. All left single-molecule signals (integrated photons per particle derived from SLIMfast) were put into a histogram for mono-/bimodal Gaussian fitting. To compare intensity distributions from different areas in the field of view, we divided the full field of illumination into 6 equally-sized centered circular areas (A1, …, A6 in Supplementary Fig. 4). We sorted all localizations into these areas and compared them with the full field of illumination and the image splitter (QuadView, QV) window. The results are summarized in Supplementary Table 1.

**Intensity-based stoichiometry analysis in live cells.** Single-molecule localization and tracking was performed in SLIMfast. After localization, immobile particles were filtered out using DBSCAN with a search radius of 100 nm and an observation probability of 50%. Spurious background localizations were removed by applying a signal-to-noise ratio filter (22 dB). Remaining localizations were tracked with a minimum track length of >= 4 frames. Single-molecule signals (integrated photons per particle) of trajectories longer than 3 seconds were put into histograms for Gaussian fitting for monomers or bimodal Gaussian fitting for monomers and dimers, respectively. In order to obtain diffusion coefficients (D), localizations of potential dimers were separated from monomers with a threshold for photons per particle. The resulting subsets of localizations with high and low signals were

tracked again to calculate the mean squared displacement (MSD) per trajectory in SLIMfast. The final diffusion coefficients were derived by the slope of a linear fit based on the first 5 data points of each single MSD.

**smFRET dimerization analysis in live cells.** In order to obtain the relative dimerized fraction ($f_{rel}^D$) of a homodimeric receptor system by smFRET, we aimed to count the average number of monomeric receptors ($N_m$) as well as the number of receptors being part of dimers ($N_d$):

$$f_{rel}^D = \frac{N_d}{N_m + N_d}. \tag{4}$$

Since these experiments were based on dual-color labeling of the same receptor and equimolar labeling concentrations could not be precisely controlled, we needed to correct for the labeling ratio ($r_l$) between label 1 and label 2. To calculate the total number of monomers $N_m = n_m^1 + n_m^2$, we accounted for false-positive monomers detected by dimers ($n_d^{11}, n_d^{22}, n_d^{12*}$):

$$n_m^1 = n_m^{1*} - n_d^{11} - n_d^{12*}, \tag{5}$$

$$n_m^2 = n_m^{2*} - n_d^{22} - n_d^{12*}. \tag{6}$$

Here, $n_m^{1*}$ and $n_m^{2*}$ are based on the directly measured averaged number of localizations in each channel. To this end, channel registration of raw time-lapse movies was performed using projective transformation obtained from TetraSpec™ microsphere (ThermoFisher) images. Afterwards, directly excited donor and acceptor signals were localized, cluster filtered (DBSCAN; 100 nm, 50%) and tracked with SLIMfast. Trajectories longer than 10 consecutive frames were transformed back into localizations. These localizations were averaged over all frames to get final numbers for $n_m^{1*}$ and $n_m^{2*}$. The labeling ratio can be calculated by $r_l = n_m^{1*}/(n_m^{1*} + n_m^{2*})$. Furthermore, $n_d^{12*}$ is the directly measured averaged number of sensitized FRET signals by donor-excitation. Sensitized FRET signals sometimes exhibited critically low S/N ratios to be robustly tracked. Therefore, we only localized FRET signals and co-localized them afterwards with localizations of directly excited acceptor signals. Thereby, we circumvented false-positive spurious detection and loss of localizations due to poor tracking fidelity at the same time. Co-localization radius was set to 150 nm, to cope with the slight time delay of 25 ms between donor and acceptor excitation. These acceptor-co-localized FRET localizations were averaged to obtain final numbers for $n_d^{12*}$.

The total number of receptors being part of dimers was calculated by:

$$N_d = 2(n_d^{11} + n_d^{12*} + n_d^{22}). \tag{7}$$

The factor of 2 accounts for two receptors being part of one dimer. Due to the high number of false-negative dimers ($n_d^{11}$ and $n_d^{22}$), we estimated the total number of dimers $N_d$ by dividing $n_d^{12*}$ by its measurement probability:

$$P(n_d^{12*}) = 2 \cdot r_l \cdot (1 - r_l), \tag{8}$$

$$\Rightarrow N_d = 2 \frac{n_d^{12*}}{P(n_d^{12*})}. \tag{9}$$

In the ideal case of a labeling ratio of $r_l = 0.5$, one would obtain a correction factor of 2. In the next step, we needed to replace false-

positive monomers in Eqs. (5) and (6) by

$$n_d^{11} = \frac{N_d}{2} P(n_d^{11}) \text{ with } P(n_d^{11}) = 2 \cdot r_l^2, \tag{10}$$

$$n_d^{22} = \frac{N_d}{2} P(n_d^{22}) \text{ with } P(n_d^{22}) = 2 \cdot (1 - r_l)^2. \tag{11}$$

By inserting Eqs. (10–11) into Eqs. (5–6), we obtained for the total number of monomers:

$$N_m = (n_m^{1*} + n_m^{2*} - \frac{n_d^{12*}}{r_l \cdot (1 - r_l)}). \tag{12}$$

Using $c_l = r_l \cdot (1 - r_l)$ as labeling correction factor, we finally derived the following relation for the relative dimerization fraction:

$$f_{rel}^D = \frac{n_d^{12*}}{c_l (n_m^{1*} + n_m^{2*})}. \tag{13}$$

Receptor densities were obtained by the average number of localizations $n_m^{1*}$ and $n_m^{2*}$ divided by the estimated area of the cell.

**Quantitative FRET efficiency analysis.** Alternating laser excitation single-molecule FRET experiments provide three emission channels: Directly excited donor ($D_{ex}$) and acceptor ($A_{ex}$) channels ($F_{D_{ex}}^{D_{em}} + F_{A_{ex}}^{A_{em}}$) as well as a sensitized FRET channel ($F_{D_{ex}}^{A_{em}}$). Fluorescence channels were aligned using a projective transformation obtained from TetraSpec™ microsphere (ThermoFisher) images. In order to robustly localize directly excited donor and FRET signals, the FRET channel ($F_{D_{ex}}^{A_{em}}$) was merged with the donor channel ($F_{D_{ex}}^{D_{em}} + F_{D_{ex}}^{A_{em}}$). After applying a highly sensitive and selective localization algorithm[95], the respective single-molecule intensities ($I_{D_{ex}}^{D_{em}}, I_{A_{ex}}^{A_{em}}, I_{D_{ex}}^{A_{em}}$) for each channel were calculated from background-subtracted images. To this end, a Gaussian weighted sum was applied which was based on the experimental point spread function estimated by raw data. Histograms of directly excited acceptor intensities ($I_{A_{ex}}^{A_{em}}$) were fitted with a bimodal Gaussian. Signals from the lower small peak ($< 5\%$) were neglected (spurious dim signals) to filter for real acceptor signals. To detect real donor-acceptor pairs in time, the merged donor-FRET signals and the acceptor signals were co-localized based on an optimized search radius. The search radius was estimated by the mean diffusion coefficient of single receptors as well as mean localization precisions for each channel. Only these co-localizations were used to calculate the apparent FRET efficiency $E_{raw}$ by:

$$E_{raw} = \frac{I_{D_{ex}}^{A_{em}}}{I_{D_{ex}}^{D_{em}} + I_{D_{ex}}^{A_{em}}}. \tag{14}$$

As described in detail elsewhere, typical further corrections were applied in order to obtain accurate FRET efficiencies[96]. To this end, the donor-only fraction was filtered for spurious signals by fitting with a bimodal Gaussian to obtain real donor-only signals. These signals were used to calculate the donor leakage coefficient $l$ by

$$l = (I_{D_{ex}}^{A_{em}} / I_{D_{ex}}^{D_{em}}) \tag{15}$$

which experimentally averaged at -11%.

The direct excitation coefficient (Dir) of the acceptor by donor laser excitation could not be experimentally discriminated from noise and was estimated by parameter tuning to 1%. FRET signals were then corrected to obtain zero FRET intensities for co-localizing receptors that are not interacting:

$$I_{D_{ex}}^{*A_{em}} = I_{D_{ex}}^{A_{em}} - l \cdot I_{D_{ex}}^{D_{em}} - \text{Dir} \tag{16}$$

With these corrections, the crosstalk corrected proximity ratio $E_{PR}$ was calculated:

$$E_{PR} = \frac{I_{D_{ex}}^{*A_{em}}}{I_{D_{ex}}^{D_{em}} + I_{D_{ex}}^{*A_{em}}} \tag{17}$$

The correction factor $\gamma$ based on emission detection efficiency could not robustly be determined from experimental data, therefore a theoretical $\gamma$ was calculated taking the detection pathway of the microscope and the dye pair Cy3B/ATTO 643 into account:

$$\gamma = \phi_A \eta_{A_{ex}}^{A_{em}} / \phi_D \eta_{D_{ex}}^{D_{em}}, \tag{18}$$

where $\phi$ is quantum yield and $\eta$ detection efficiency. Using the manufacturer's transmission/detection spectra for all optics in the detection pathway of the respective channels and the dye spectra taken from FPbase[97] resulted in $\gamma = 0.8755$. With the same approach, the Cy3B cross-emission derived from this calculation was 11% (Supplementary Fig. 7a), which matched the experimental results. Background- and bleed through corrected $E_{PR}$ was then corrected for $\gamma$, to obtain accurate FRET efficiencies $E$:

$$E = \frac{E_{PR}}{(\gamma - (\gamma - 1)E_{PR})}. \tag{19}$$

Accurate FRET efficiencies were histogrammed and the peak value was converted into distance by

$$E = \frac{1}{1 + \left(\frac{r}{R_0}\right)^6} \tag{20}$$

with $R_0^{Cy3B/AT643} = 6.441$ nm.

Before acquisition, the laser power was adjusted to get similar intensities for donor and acceptor monomers and the apparent stoichiometry $S_{raw}$ was used for E-S plots:

$$S_{raw} = \frac{I_{D_{ex}}^{D_{em}} + I_{D_{ex}}^{A_{em}}}{I_{D_{ex}}^{D_{em}} + I_{D_{ex}}^{A_{em}} + I_{A_{ex}}^{A_{em}}} \tag{21}$$

**Long-term localization and tracking.** Since SLIMfast showed performance issues for tracking and visualization of datasets ≥ 2000 frames, we decided to use TrackMate 7.10.2. in FIJI[89,93,94]. Single-molecule signals were detected and localized using a difference of Gaussian (DoG) filter with median filtering and subsequent subpixel accuracy localization. The particle radius was set 0.2 μm (Cy3B) or 0.22 μm (ATTO 655) and the quality threshold was set by visual inspection of detected particles. Tracking was done using the Simple LAP tracking algorithm with a linking max gap of 500 nm and maximum 2 frames for gap closure. Gap closure max. distance was set to 500 nm as well. All trajectories were filtered for length ≥ 10 frames. For visualization of trajectories we rendered all filtered trajectories by built-in functions in TrackMate. Trajectories were color-coded by time [0–max], confinement ratio [0, 0.3] or mean speed [2 μm/s, 6 μm/s]. The confinement ratio in TrackMate is defined as the net-displacement divided by the total-distance. It is unitless and ranges from 0 to 1. Values close to 0 indicate confinement, where the molecule stays close to its starting point. Values close to 1 indicate the molecule travels along a line with a constant orientation (active/directed movement). For combined SIM and single-molecule tracking experiments, the SMT dataset was rescaled within FIJI to match the pixel size of the SIM dataset before using TrackMate. Image registration of both multiple channels was achieved by imaging 100 nm TetraSpec™ microspheres (Thermo-Fisher) and using a FIJI plugin for descriptor-based registration[98] with subpixel fitting and affine transformation.

**DNA-PAINT analysis.** Experiments with Gattaquant nanorulers were analyzed with Picasso 0.6.1[99]. Raw data was split into each channel and channel registration was performed using descriptor-based registration[98] with subpixel fitting and affine transformation using again images of 100 nm TetraSpec™ microspheres (ThermoFisher). Each channel is processed by Picasso localizer using a box size of 7 pixels, a min net. Gradient of 10,000 for thresholding and maximum likelihood estimation with default settings for single emitter localization. Datasets were filtered by localization precision $\leq 0.1$ pixel (10 nm) und drift was removed by Picasso Render using the RCC algorithm with a segmentation of 500 frames. After picking nanoruler structures in Picasso Render, all picked structures were averaged in Picasso Average using 10 iterations and an oversampling of 25×. Averaged images were exported to Matlab for fitting with a Gaussian mixture model with three populations.

## SIM reconstruction and post processing

Reconstruction of super-resolved SIM images was done by fairSIM (https://www.fairsim.org/) using an estimated optical transfer function (OTF) and standard Wiener filtering[53] followed by Hessian denoising[54]. A good reconstruction requires robust detection of reconstruction parameters from raw data. Since high-speed SIM data had typically low signal-to-noise levels, we averaged 10 time points (90 SIM frames) before parameter estimation by fairSIM. Data loaded into fairSIM was always corrected for background and pattern orientation-dependent differences in illumination intensity before any further processing. The OTF in fairSIM was estimated by giving the emission wavelength (green channel: 525 nm, orange channel: 585 nm, red channel: 680 nm) and the effective numerical aperture NA = 1.33 considering the refractive index mismatch at the glass/medium interface. The OTF dampening parameter was set to the default value $a = 0.3$. OTF attenuation was deactivated. The reconstruction settings were set to default values: Filter type: Wiener filter; Wiener parameter: 0.05; Apodization cutoff: 2; apodization bend: 0.9. Time-lapse data was reconstructed in time-lapse mode generating a super-resolved dataset as well as a diffraction-limited widefield representation. Super-resolved datasets further processed by Hessian denoising using default parameters ($\mu = 150$, $\sigma = 1$). Channel registration for multi-color SIM acquisitions was achieved by imaging 100 nm TetraSpec™ microspheres (ThermoFisher) and using a FIJI plugin for descriptor-based registration with subpixel fitting and affine transformation[98].

## WEKA segmentation of actin cortex

All following steps were performed in FIJI[89]. After Hessian denoising of SIM reconstructions, we applied gamma correction (parameter = 0.5) to balance very bright signals from actin patches and dim signals from fine filaments. Then, we performed unsharp masking (radius = 1 pixel, mask weight = 0.6) to smooth data but preserving edges. This dataset was upscaled by a factor of two to improve resolution for WEKA segmentation. We used the FIJI plugin Trainable WEKA Segmentation v3.3.4[55]. Robust segmentation required four classes: 1. Background outside cell, 2. Background inside cell, 3. Fine filaments (actin cortex) and 4. Bright patches (focal adhesions). After training, we exported probability maps for class 3 (actin cortex) as 16-bit time-lapse stacks. Then, we applied a threshold to these maps using the OTSU method to obtain a binary representation of the fine actin cortex. The last steps were skeletonization to increase resolution and down-sampling to original resolution to obtain smooth high-resolution filament networks.

## Determination of actin cortex corral sizes

All processing steps were performed in FIJI[89]. The first time point of the WEKA segmentation was used to analyze mean corral sizes. We applied an automatic threshold using the default method and inverted the image to obtain white corrals on black network. The sizes of the corrals were analyzed with the built-in particle analyzer tool in FIJI. Maximum particle size was set to 0.25 $\mu m^2$. Statistics of 3148 areas were exported as csv to plot size distribution in Matlab.

## Co-localization and co-tracking analysis of endosomal FYVE and TpoR signals

Tracking and co-localization analysis of endosomal FYVE and TpoR signals was accomplished using Imaris 9.5 (Bitplane). Single-molecule datasets were time-averaged with a window of 3 s (90 frames). StayGold-2xFYVE was imaged every 3 s. Original SIM datasets were averaged to obtain diffraction-limited widefield images. SIM reconstructions of FYVE-signals showed quite some artifacts most likely due to fast dynamics of the diffraction-limited spots during SIM acquisitions. Nonetheless, higher resolution by SIM would not improve co-localization and co-tracking analysis in this case. We used the spots detection routine within Imaris: We localized and tracked slow-moving FYVE and TpoR signals with a max. step length of 300 nm/frame. By exclusion of very dim signals and filtering for long trajectories ( > 10 frames (30 s)), we could robustly follow single signals over time. Co-localization was based on defining a maximum distance of 400 nm between single spots. Final filtering is based on a second tracking step of co-localized signals only with a track length $\geq 60$ frames (180 s).

## Resolution estimation by decorrelation analysis

To investigate SIM reconstruction performance with and without using a beam-shaping device, we applied parameter-free decorrelation analysis in Matlab (R2022b, Mathworks) to determine local image resolution enhancement by SIM[100]. Briefly, pseudo diffraction limited as well as SIM reconstruction images were Fourier transformed after standard edge apodization to suppress high-frequency artefacts. The Fourier transform was normalized and the input image $I(k)$ and its normalized version $I_n(k)$ are then cross-correlated in Fourier space using Pearson correlation to obtain a single value between 0 and 1. These operations were repeated, but the normalized Fourier transform was filtered additionally by a binary circular mask of radius $r \in [0,1]$ expressed in normalized frequencies resulting in the final decorrelation function $d(r)$ (further details in[100]). The input image was also high-pass filtered (from weak to very strong filtering by $N_G$ Gaussian filters) to attenuate the energy of low frequencies. For each filtered image, $d(r)$ was computed and the peak positions were extracted. The cut-off frequency $k_c$ was estimated by the maximum spatial frequency of each round of high-pass filtering: $k_c = \max[r_{0,\ldots,r_{N_G}}]$. Final resolution was calculated from the cut-off frequency by $2 \times pixel\ size/k_c$. We imaged 100 nm sized TetraSpec™ microspheres as described above for channel registration to analyze overall as well as tiled local resolution enhancement by decorrelation analysis (Supplementary Fig. 9). SIM data was acquired for each channel (488 nm, 560 nm, 642 nm) at six different sample positions. Pseudo diffraction-limited images as well as SIM reconstructions were calculated as described above. Results can be compared with direct resolution estimation by PSF fitting (Supplementary Fig. 10).

## Experimental conditions

All experimental conditions are summarized in Supplementary Table 3.

## Statistics and Reproducibility

Each experiment was repeated at least twice using independent experimental samples. Statistical tests were performed using two-sample Kolmogorov–Smirnov tests. P-values are given in Fig. 6g and Supplementary Figs. 6i, 6l and 15i. Box charts with data points were created using Origin 9.0 with standard settings: Boxes show inter-quartile range (25–75%) and median (line), whiskers show outliers. Additionally, minimum and maximum (-), 1-99% (x) and mean (□) are

indicated. Following histograms or line plots (Figs. 1g, 2d, 2f, 3i, Supplementary Figs. 2b, 2d, 3b, 3d, 3f, 3h, 4d, 4g, 5f, 5g, 6d, 6f, 6j, 6k, 7b, 7c, 10b, 14a, 15g) were fitted using Matlab R2022b or Origin 9.0 by Gaussian mixture models and 1, 2 or 3 populations as indicated in each figure. Shown standard deviations ($\sigma$) and errors are based on Gaussian fits with 95% confidence interval.

## Reporting summary

Further information on research design is available in the Nature Portfolio Reporting Summary linked to this article.

## Data availability

The microscopy data generated in this study have been deposited in the OsnaData repository under accession code: https://doi.org/10.26249/FK2/V8HWWH. All plasmids from this study were deposited at Addgene (https://www.addgene.org/, IDs 222944-222949). Source data are provided with this paper.

## Code availability

The Matlab R2013a software SLIMfast (Version 16h) is available via https://doi.org/10.5281/zenodo.3588413 and the Matlab R2020b simulation code SPT_Simulator V1.0.0 can be found under https://doi.org/10.5281/zenodo.11203941. The Matlab R2022b script for FRET efficiency analysis accompanied with a demo dataset is provided as Supplementary Software with this paper.

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

## Acknowledgements

We thank A. Budke-Gieseking, G. Hikade, H. Kenneweg and Wladislaw Kohl for technical support. We thank Janelia Materials for providing HTL-JFX549 and HTL-JFX646 and Kai Johnsson for providing HTL-MaP555. We thank Atsushi Miyawaki for depositing StayGold (Addgene #185823) and Harald Stenmark for depositing FYVE (Addgene #140047). We thank Isabelle Watrinet for providing tris-NTA functionalized glass coverslips. This work was funded by the Deutsche Forschungsgemeinschaft (DFG, German Research Foundation) – SFB 944;1557 - 180879236; 467522186 to J.P. (SFB 944/P8, SFB 944/Z1 & SFB 1557/P13) and R.K. (SFB 944/Z1 & SFB 1557/Z2).

## Author contributions

Conceptualization and Methodology: H.W., J.P. and R.K. R.K. built the microscope and developed simulation software. H.W., J.E. and R.K. performed all experiments and analyzed data. H.W., J.P. and C.P.R. designed single-molecule FRET experiments and data evaluation. Funding acquisition, project administration & supervision: J.P. and R.K. Writing—original draft: H.W., J.P., and R.K. Writing—review and editing: all authors.

## Funding

## Competing interests

The authors declare no competing interests.
