## [Peer Review File · Nature Communications]

Reviewers' Comments:

Reviewer #1:

Remarks to the Author:

The authors have used a combination of flat-field TIR illumination with SIM to obtain single molecule fluorescence data at temporal resolutions that can provide information about receptor dynamics. To establish this, they used the TpoR as their model system. Their methodology is robust and builds upon their previous workplan (Wilmes et al, 2020) where they used a method of co-tracking the receptors to establish that homodimerization is a prerequisite for signaling activation by TpoR, EpoR and GHR. The present article although robust in its spatio-temporal resolution data does not provide insights into the physiological receptor dynamics.

Major comments

1. The authors observed that 12% of the receptors dimerize in the presence of the tdALFAnb. Is this related to the affinity of the nb for the receptor? Would these percentages change if the natural ligand Tpo was used?
2. Given that TpoR is known to signal through multiple dimeric interface and the low dimerization event observed in their system, it is good to check the extent of receptor mediated signaling observed in their set-up.
3. It has been shown in Fig. 2 that upon bleaching ATTO643 labelled TpoR in the presence of dimerization inducing tdALFAnb, the fluorescence drops to half. Again, the same could be checked for the natural ligand Tpo.
4. In the line number 215-216, the authors mention that co-tracking was limited by termination due to crossing of molecules. Could they explain this point clearly.
5. Continuing with the above point, have the authors observed an event of multimerization (tetramers etc) of TpoR through smFRET?
6. Did the authors observe clusters of TpoR in endocytic pits and what was the size of these clusters and the endocytic pits? This could greatly help define the results in Fig. 4.
7. How do the authors induce endocytosis of the receptor in the absence of its ligand in Fig. 4.
8. Was crowding of the receptors sufficient for dimerization and endocytosis? Again, receptor signaling data should be included.
9. The experiments in Fig. 4 must be corroborated with a TpoR dimerization inducer – ligand Tpo and the cross-linking nanobody. It will be interesting to observe for similarities or differences in the track lengths or rate of endocytosis for the two dimerization inducing agents.
10. It is hard to understand how single receptor molecules would be endocytosed (line 238-239). Could there possibly be a different explanation for the single channel events that disappear?
11. Line 300-304, the authors mention that the TpoR close to MSK network decreases gradually in its intensity possibly due to endocytosis. This needs to be verified using an endocytosis marker.
12. Further in line 300-304, they state that under these conditions, a step-wise decrease in the fluorescence intensity due to photobleaching was not observed. Is it possible for the authors to correlate the fluorescence with the distance of the fluorophore to get an estimate of the size of the invagination.
13. The actin meshwork data is interesting and demonstrates again the technical advancement in combination with the AI driven analysis in identifying thin actin network dynamics. But it does not add anything substantial to the TpoR story.

Reviewer #2:

Remarks to the Author:

In this work by Winkelmann et al. the authors develop a new microscope illumination system that produces an extremely homogeneous total internal reflection field that can be rapidly switched to a structured illumination mode. This allows for nice quantitative imaging of single molecules, FRET signals, and super-resolved cellular structures in live cells in several different colors. The illumination homogeneity of TIRF has been a long standing problem. This is due to many factors including single angle illumination, polarization of the beam, interference patterns generated by

the optical components, beam alignment issues, and scattering in the objective. The very impressive flat fields generate in the paper by a fairly simple optical system is commendable. The thoughtful and quantitative analysis is also quite nice and impressive. These development--if widely implemented-- would certainly improve the quality of imaging in labs that use these methods. I think overall this is a nicely done study.

I have two comments the authors might consider to improve the manuscript.

1. The title and abstract of the paper "Correlative single..." Leads one to assume that the paper is mostly centered around this concept of matching single molecule imaging to SIM. However, the majority of the figures leverage the homogeneous field to quantitate single molecule data. Only the last figure and a very limited set of experiments fully address the correlative possibilities. I would propose a new title and abstract that better encompasses the majority of the work in the paper. Otherwise, I would suggest an expansion of the correlative experiments and analysis and a more robust emphasis on the advantages of that type of dual correlative imaging.

2. It is difficult to compare/judge the improvements in this system to other published and widely-available systems that use (for example) ring-TIRF illumination and/or multibeam TIRF-SIM. Can the authors better present a quantitative comparison to other imaging systems such that the reader can evaluate the relative improvements and need for this system as a whole? For example, this was nicely done in the original Stehr et al. paper on flat top TIRF illumination that the authors reference in the paper.

Reviewer #3:

Remarks to the Author:

Uniform illumination plays a crucial role in quantitative fluorescence microscopy techniques. While flat-field illumination is commonly used in single-molecule localization imaging systems, it has yet to be applied in structured illumination microscopy (SIM). To address this gap, the authors combined a piShaper with a spatial light modulator to achieve multi-angular homogenous total internal reflection fluorescence (TIRF) illumination for correlative single-molecule and structured illumination microscopy. This innovative combination holds promise as a valuable tool for investigating molecular organizations and dynamics.

A major concern raised in this manuscript is using piShaper as a preferred tool for achieving flat-top illumination in single-molecule approaches and its compatibility with TIRF illumination. However, it is important to note that piShaper achieves uniform illumination by redistributing light intensity, which simultaneously distorts the wavefront. This distortion could hinder the generation of high-quality interference patterns required for successful SIM imaging. It is plausible that this inherent limitation of piShaper is the primary reason why it has not been utilized in SIM, despite its popularity in the single-molecule imaging community.

Another key observation discussed in this manuscript pertains to the field of view in SIM, which appears to be small despite the large region of homogenous illumination. This suggests that only the central region of the sample provides satisfactory interference patterns required for SIM reconstruction. Essentially, this approach is not significantly different from the conventional practice of using the center region of a Gaussian illumination in SIM. To address this limitation, it is recommended that the authors conduct a quantitative investigation and experimental demonstration to evaluate the performance of piShaper in SIM illumination and reconstruction. Such an assessment would provide solid evidence to support the potential utility of piShaper in enhancing SIM techniques.

Response to the reviewers

Reviewer #1:

The authors have used a combination of flat-field TIR illumination with SIM to obtain single molecule fluorescence data at temporal resolutions that can provide information about receptor dynamics. To establish this, they used the TpoR as their model system. Their methodology is robust and builds upon their previous workplan (Wilmes et al, 2020) where they used a method of co-tracking the receptors to establish that homodimerization is a prerequisite for signaling activation by TpoR, EpoR and GHR. The present article although robust in its spatio-temporal resolution data does not provide insights into the physiological receptor dynamics.

Response: We thank the reviewer for the constructive assessment of our work and we appreciate the helpful suggestions. We agree that this work does not provide fundamentally new insights into TpoR signaling, but rather uses this system as a model for testing the performance of a new microscope and demonstrating its capabilities. With this focus on microscopy development, addressing specific biological questions is beyond the scope of our study; yet, we agree with the reviewer that application of imaging techniques under physiologically more relevant conditions would increase the impact of our work. We therefore performed additional experiments with the natural ligand and with active TpoR signaling complexes, with particular focus on receptor stoichiometries and endocytosis, which has not yet been addressed previously. These results were included into the revised manuscript (Fig. 6 and Supplementary Fig. S6). For a more thorough intensity-based analyses of receptor stoichiometries, we have devised a simulator to generate realistic images (Supplementary Fig. S4). Based on single molecule intensity analysis of simulated images, more meaningful interpretation of the experimental data has been achieved.

Major comments

1. The authors observed that 12% of the receptors dimerize in the presence of the tdALFAnb. Is this related to the affinity of the nb for the receptor? Would these percentages change if the natural ligand Tpo was used?

Response: We have employed the TpoR system in the absence of JAK2 and the artificial dimerizer as a model system for a monomer-dimer system unbiased by downstream signaling events. We have previously shown that the presence of JAK2 already induces weak TpoR dimerization (Wilmes et al., 2020), and wanted to avoid such bias in these experiments. These interactions mediated by the pseudokinase domains of JAK2 cooperate with dimerization induced by the natural ligand thrombopoietin (TPO). We therefore used the artificial dimerizer at an optimized concentration to have a clean, efficient switch from monomers to dimers.

Inspired by the reviewer's comments, we have performed and included additional experiments with TpoR in the presence of JAK2 and dimerized by its natural ligand thrombopoietin (TPO). In these experiments, we used JAK2 lacking the tyrosine kinase domain to allow efficient TPO-induced dimerization while avoiding potential bias caused by downstream signal activation (new Figure S6). Furthermore, we performed long-term SMI of activated TpoR signaling complexes in the presence of full-length JAK2 and included intensity analyses (new Fig. 6). Strikingly, we find a very similar monomer/dimer switch upon stimulating with TPO, with negligible higher-order oligomers being induced.

2. Given that TpoR is known to signal through multiple dimeric interface and the low dimerization event observed in their system, it is good to check the extent of receptor mediated signaling observed in their set-up.

Response: As pointed out above, the artificial dimerizer was used as a controlled monomer/dimer switch under conditions that rule out bias caused by inducing downstream signaling. Rather, we now quantified by intensity analysis TPO-induced dimerization under conditions we have previously demonstrated to mimic the formation of active signaling complexes. These results were included as Fig. S6 into the revised manuscript and referred to the previously demonstrated signaling activity of this reconstituted system.

3. It has been shown in Fig. 2 that upon bleaching ATTO643 labelled TpoR in the presence of dimerization inducing tdALFAnb, the fluorescence drops to half. Again, the same could be checked for the natural ligand Tpo.

Response: We have included the corresponding analysis for TPO-induced TpoR dimers into Fig. S6g,h.

4. In the line number 215-216, the authors mention that co-tracking was limited by termination due to crossing of molecules. Could they explain this point clearly.

Response: This statement refers to single molecule tracking rather co-tracking, and refers to a very common and well-known feature: If two particles temporarily come into proximities closer than their frame-by-frame jump distance, faithful tracking cannot be warranted and therefore trajectories are aborted. Since this is common knowledge in the field, we refrained from further explanations in the manuscript.

5. Continuing with the above point, have the authors observed an event of multimerization (tetramers etc) of TpoR through smFRET?

Response: Neither single particle intensity analysis nor smFRET indicated TpoR multimerization. However, these analyses were limited to signaling-inactive TpoR as explained above. Moreover, we omitted the immobile fraction in these analyses, which we now clearly show to be related to endocytosis (see below). In the revised version, we have extended our analysis to active, TPO-induced TpoR dimers and included a comparison of the mobile and the immobile fraction, which we have now confirmed to be related to endosomal uptake. While we see a clear bias towards TpoR dimers in the immobile fraction, higher oligomers are extremely scarce. These results have been summarized in the new Fig. 6.

6. Did the authors observe clusters of TpoR in endocytic pits and what was the size of these clusters and the endocytic pits? This could greatly help define the results in Fig. 4.

Response: See response to question no. 5.

7. How do the authors induce endocytosis of the receptor in the absence of its ligand in Fig. 4.

Response: Class I/II cytokine receptors are rapidly endocytosed even in the resting state, with diverse uptake mechanisms being implicated. This can explain the relatively low cell surface expression even under conditions of overall high overexpression. In some cases, receptor uptake kinetics has been shown to be largely independent on receptor stimulation. We have included some more explanations and references on cytokine receptor endocytosis into the revised manuscript.

8. Was crowding of the receptors sufficient for dimerization and endocytosis? Again, receptor signaling data should be included.

Response: We have previously shown that the TpoR cell surface density used for single molecule experiments is close to the physiological expression levels, and that potent signaling can be stimulated under these conditions (Wilmes et al., 2020). There is no indication that the relative probability of endocytosis depends on receptor cell surface density

or that crowding/clustering would be needed. As pointed out above, we have previously confirmed high signaling activity of the fully reconstituted TpoR/JAK2 system that we here apply for more detailed analysis of endocytic trafficking.

9. The experiments in Fig. 4 must be corroborated with a TpoR dimerization inducer – ligand Tpo and the cross-linking nanobody. It will be interesting to observe for similarities or differences in the track lengths or rate of endocytosis for the two dimerization inducing agents.

Response: We have performed long-term single molecule tracking experiments with TPO-stimulated TpoR and included these experiments and analyses into Fig. 6 and Supplementary Fig. S14. We complemented these studies with simulated imaging data, which enabled more reliable interpretation.

10. It is hard to understand how single receptor molecules would be endocytosed (line 238-239). Could there possibly be a different explanation for the single channel events that disappear?

Response: See response to question no. 7.

11. Line 300-304, the authors mention that the TpoR close to MSK network decreases gradually in its intensity possibly due to endocytosis. This needs to be verified using an endocytosis marker.

Response: We have performed additional experiments to co-localize immobile TpoR with a generic endosomal marker (tandem-FYVE binding phosphatidylinositol 3-phosphate) and found a striking, time-dependent correlation. These results were included as Fig. 6 into the revised manuscript.

12. Further in line 300-304, they state that under these conditions, a step-wise decrease in the fluorescence intensity due to photobleaching was not observed. Is it possible for the authors to correlate the fluorescence with the distance of the fluorophore to get an estimate of the size of the invagination.

Response: This is a very interesting suggestion. For such analyses, however, we would need to estimate the number of labeled molecules in these compartments, which cannot be directly derived by the intensity since fluorescence intensity additionally depends on the axial position within the evanescent excitation field. Since the original analysis was based on time-lapse mean intensity projections of the TpoR channel, we added a single-particle analysis with full temporal resolution to the revised manuscript (Supplementary Fig. S13 and Movie S16). These results suggest that endocytic TpoR signals in the resting state of the cell can consist of more than a single labeled TpoR. However, assuming a fixed number of labeled molecules and an evanescent field with a fixed penetration depth (Supplementary Fig. S4e), one could try to estimate some distances. Nonetheless, our simulations and stoichiometry analysis (Supplementary Fig. S6) suggest plasma membrane ripples with axial position variations in the range of 20-30 nm which should be considered in such an analysis. Furthermore, one would need to capture the full process from the start of endocytosis to invagination. Since we think that such analyses are not highly relevant to the conclusions of this work, we abstained from performing such calculations here.

13. The actin meshwork data is interesting and demonstrates again the technical advancement in combination with the AI driven analysis in identifying thin actin network dynamics. But it does not add anything substantial to the TpoR story.

Response: As pointed out above and already stressed throughout the manuscript, the focus of this work is to demonstrate the diverse capabilities of a new microscope setup. We have

used the TpoR system to coherently characterize the performance of the different features and to highlight the new possibilities to tackle exciting questions in receptor biology. With the long-term tracking and endocytosis analysis, which we have extended in the revised version, we could shed new insights into the spatiotemporal dynamics of TpoR in resting and activated state. However, following up in more detail all the possibilities that we have demonstrated in these experiments will be clearly beyond the scope of this paper.

Reviewer #2:

In this work by Winkelmann et al. the authors develop a new microscope illumination system that produces an extremely homogeneous total internal reflection field that can be rapidly switched to a structured illumination mode. This allows for nice quantitative imaging of single molecules, FRET signals, and super-resolved cellular structures in live cells in several different colors. The illumination homogeneity of TIRF has been a long standing problem. This is due to many factors including single angle illumination, polarization of the beam, interference patterns generated by the optical components, beam alignment issues, and scattering in the objective. The very impressive flat fields generate in the paper by a fairly simple optical system is commendable. The thoughtful and quantitative analysis is also quite nice and impressive. These development--if widely implemented-- would certainly improve the quality of imaging in labs that use these methods. I think overall this is a nicely done study.

Response: We thank the reviewer for the positive feedback. We absolutely agree that multi-color homogenous illumination under stringent TIR conditions is very challenging. As discussed in detail in the manuscript, we agree that our approach is fairly simple since it is based on well-known two-dimensional SIM setups and open-source imaging software. We want to stress that especially multi-color live-cell single-molecule tracking requires optimal wavelength-dependent TIR conditions for each channel to obtain robust quantitative data. Therefore, we mainly focused on demonstrating the performance of our setup with regard to intensity-based stoichiometry analysis, long-term single-molecule tracking and analysis as well as quantitative single-molecule FRET of diffusing receptor complexes in living cells. The combination with SIM opens new exciting possibilities to correlate quantitative single-molecule imaging with super-resolution of cellular dynamics in living cells.

I have two comments the authors might consider to improve the manuscript.

1. The title and abstract of the paper “Correlative single...” Leads one to assume that the paper is mostly centered around this concept of matching single molecule imaging to SIM. However, the majority of the figures leverage the homogeneous field to quantitate single molecule data. Only the last figure and a very limited set of experiments fully address the correlative possibilities. I would propose a new title and abstract that better encompasses the majority of the work in the paper. Otherwise, I would suggest an expansion of the correlative experiments and analysis and a more robust emphasis on the advantages of that type of dual correlative imaging.

Response: As mentioned above, we focused on demonstrating robust single-molecule imaging techniques (stoichiometry, long-term multi-color tracking, single-molecule FRET) which rely on optimal illumination conditions. This was the most challenging part of the study and therefore covers the main part of the story. Nonetheless, we would like to keep to title “Correlative single-molecule...” as it is, since we think that the combination of quantitative single-molecule imaging and live-cell super-resolution makes this microscope design in combination with tailored analysis workflows particularly unique and very powerful. We demonstrated the combination of single-molecule imaging and TIRF-SIM by imaging the actin-cortex and the thrombopoietin receptor TpoR in the resting state of the cell (Fig. 5) and

after stimulation with its natural ligand thrombopoietin (new Fig. 6 and new Supplementary Figure S14, new Movies S18-20). We further analyzed endocytic TpoR by co-localizing the single-molecule channel with TIRF imaging of tandem-FYVE-StayGold used as an endosomal PIP3 marker after receptor stimulation (Fig. 6a-d, Supplementary Movie S17). TpoR dynamics on the plasma membrane and in endosomes was correlated with actin corral dynamics under physiological conditions (Fig. 6e-g, Supplementary Figure S14, Supplementary movie S18-20). Furthermore, we also demonstrated the possibility of intracellular single-molecule tracking of the mitochondrial translocase TOMM20 and correlated the dynamics with super-resolution of the outer mitochondrial membrane (Fig. S12). We think, that these experiments demonstrate the full potential of combining single-molecule imaging and super-resolution SIM in living cells.

2. It is difficult to compare/judge the improvements in this system to other published and widely-available systems that use (for example) ring-TIRF illumination and/or multibeam TIRF-SIM. Can the authors better present a quantitative comparison to other imaging systems such that the reader can evaluate the relative improvements and need for this system as a whole? For example, this was nicely done in the original Stehr et al. paper on flat top TIRF illumination that the authors reference in the paper.

Response: We agree that a quantitative performance comparison of different available illumination modalities of our setup would be very helpful for the reader. We therefore imaged a Texas Red dye solution under all available imaging conditions (1-beam TIRF, single angle SI-TIRF, 2-angles SI-TIRF and 3-angles SI-TIRF) with and without π Shaper, analyzed line profiles and summarized the results in Supplementary Figures S2c-f. Here, we used the full camera chip and demonstrate performance of the full field of illumination. We see that using multi-angular TIR illumination (2 or 3 angles) reduces robustly the occurrence of interference fringes as shown in Fig. S2c, e. We followed the suggestion of the reviewer referring to Stehr et al. and conducted MaP555-HTL PAINTing of densely coated reHaloTagF surfaces (same experiment as in Fig. 1d-g) to analyze single-molecule intensity distributions for different centric ring region of interests (ROIs) covering the same area (number of molecules) of the field of view (new Supplementary Fig. S3 and Table 4). We analyzed four conditions (1-beam TIRF and 3-angle SI-TIRF each w/ and w/o π Shaper) and found strong differences between Gaussian and flat-top illumination basically reproducing the results of Stehr et al. Our multi-angular SI-TIR illumination performs slightly better than using a single beam flat-top TIRF design (Supplementary Figure S3c,e vs. S3g,h) at least in our setup. We see that the π Shaper drastically improves intensity analysis. Even for the smaller field of view of the QuadView, we see that the main peak does not shift between area A1, A2 and A3 and the distributions are narrower (cp. Table 3 in the methods section). As discussed in the manuscript in detail, we would like to point out again that our design does not out-perform other approaches for homogenous TIR-illumination. We implemented our strategy to establish real-time multi-color single-molecule tracking and ALEX-smFRET which require precise high-speed control of evanescent field's penetration depth for multiple channels.

Reviewer #3:

Uniform illumination plays a crucial role in quantitative fluorescence microscopy techniques. While flat-field illumination is commonly used in single-molecule localization imaging systems, it has yet to be applied in structured illumination microscopy (SIM). To address this gap, the authors combined a π Shaper with a spatial light modulator to achieve multi-angular homogenous total internal reflection fluorescence (TIRF) illumination for correlative single-molecule and structured illumination microscopy. This innovative combination holds promise as a valuable tool for investigating molecular organizations and dynamics.

A major concern raised in this manuscript is using piShaper as a preferred tool for achieving flat-top illumination in single-molecule approaches and its compatibility with TIRF illumination. However, it is important to note that piShaper achieves uniform illumination by redistributing light intensity, which simultaneously distorts the wavefront. This distortion could hinder the generation of high-quality interference patterns required for successful SIM imaging. It is plausible that this inherent limitation of piShaper is the primary reason why it has not been utilized in SIM, despite its popularity in the single-molecule imaging community.

Response: We agree that wave front distortion would greatly affect the quality of linear SI patterns. AdlOptica's π Shaper is achromatic, and the manufacturer has taken special care to ensure that only very low wave front aberrations occur due to beam shaping. A key design feature of the π Shaper optical system is that it consists of two optical components and, by using special optical surfaces, a controlled wave front transformation takes place in the space between them, thereby achieving the necessary intensity redistribution. Another important feature of the π Shaper optics is that a beam passing through the π Shaper has no or negligible wave aberration for practical applications, i.e. all beams of the input beam passing through the optical system have the same path length; this condition is very important for practical applications as it guarantees that no unwanted interference fringes occur and that the intensity profile of the result is maintained over a long distance after the π Shaper. The conditions of transformation of the intensity distribution and zero or negligible wave aberration are fulfilled simultaneously for a specific spectral range, so that the optical system of the achromatic π Shaper functions in the same way at every wavelength of this spectral range - this is achieved by manufacturing the lenses from different materials with different dispersion and using additional lenses.

Another key observation discussed in this manuscript pertains to the field of view in SIM, which appears to be small despite the large region of homogenous illumination. This suggests that only the central region of the sample provides satisfactory interference patterns required for SIM reconstruction. Essentially, this approach is not significantly different from the conventional practice of using the center region of a Gaussian illumination in SIM. To address this limitation, it is recommended that the authors conduct a quantitative investigation and experimental demonstration to evaluate the performance of piShaper in SIM illumination and reconstruction. Such an assessment would provide solid evidence to support the potential utility of piShaper in enhancing SIM techniques.

Response: We thank the reviewer for the comment. We are using the same field of view (FoV) for single-molecule imaging as for SIM. Due to the fact that our system is optimized for multi-color high-speed imaging using no mechanical filter-wheels, we have introduced four-channel image-splitters for each camera limiting the FoV to one-quadrant of the detector. The circular field of illumination (Fol) is tuned to the rectangular FoV of the image-splitter. Indeed, we have chosen a quite small FoV of $25 \times 25 \mu\text{m}^2$ which is limited by the EMCCD camera which has a quadrant of 256×256 pixels each pixel having a size of 101.5 nm for single-molecule imaging. The small Fol allows for high-intensities required for single-molecule imaging. We agree that a much bigger field of view would be very interesting for SIM, but would limit speed by introducing longer camera readout times. Our system can be easily adjusted to a field of view of $40 \times 40 \mu\text{m}^2$ by removing the additional $1.6\times$ magnification after the objective lens.

We want to clarify that we do not only use the central spot of our Fol for acquisition. Our FoV is covering about 32% of the Fol considering a circular illumination and a rectangular FoV (cp. Supplementary Fig. S9a, d). In order to demonstrate SIM performance with and without using a π Shaper, we have imaged 100 nm tetraspec microspheres and analyzed the resolution by decorrelation analysis (Descloux et al., *Nat. Meth.*, **2019**) for each color globally

as well as locally using small tiles. We summarized the results in Supplementary Figure S9. Our analysis clearly shows that there is no impact on resolution by introducing a π Shaper to a SIM setup.

Reviewers' Comments:

Reviewer #1:

Remarks to the Author:

The authors have addressed the key concern by looking at the TpoR dynamics in the physiological TpoR-TPO signalling. This article now emphasizes both the development of an advanced microscopy technique as well as the relevance to physiological processes - primarily shown through the dimerization and surface kinetics of TpoR. As such, this will be useful to a broad audience interested in microscopy/optics as well as membrane receptor biology. I recommend this article for publication in Nature Communication.

Reviewer #2:

Remarks to the Author:

I was fairly supportive of the original manuscript with two major questions. Both were adequately addressed in the revised version of the manuscript. I have no further comments.

Reviewer #3:

Remarks to the Author:

It would be great if the authors could provide a SIM image using piShaper with a larger field of view, such as 40-50 μm . One color is sufficient. Apart from that, the authors have addressed my concerns.

Response to the reviewers

We again thank the reviewers for the constructive assessment of our work and their positive feedback.

Reviewer #1 (Remarks to the Author):

The authors have addressed the key concern by looking at the TpoR dynamics in the physiological TpoR-TPO signalling. This article now emphasizes both the development of an advanced microscopy technique as well as the relevance to physiological processes - primarily shown through the dimerization and surface kinetics of TpoR. As such, this will be useful to a broad audience interested in microcopy/optics as well as membrane receptor biology. I recommend this article for publication in Nature Communication.

Reviewer #1 (Remarks on code availability):

Not applicable to me.

Reviewer #2 (Remarks to the Author):

I was fairly supportive of the original manuscript with two major questions. Both were adequately addressed in the revised version of the manuscript. I have no further comments.

Reviewer #3 (Remarks to the Author):

It would be great if the authors could provide a SIM image using piShaper with a larger field of view, such as 40-50 μm . One color is sufficient. Apart from that, the authors have addressed my concerns.

Response: We added TIRF-SIM images with a larger field of view of three representing HeLa-cells expressing LifeAct-StayGold to demonstrate a super-resolved actin-cortex in each cell (Supplementary Fig. 11). For these experiments, we removed the additional 1.6x magnification which is only required for single-molecule imaging to achieve Nyquist-Shannon sampling on the EMCCD. TIRF-SIM images were acquired with the CMOS camera without pixel binning (600 x 600 pixels) resulting in a field of view of 38.4 x 38.4 μm^2 . This is the maximum field of view using image splitter optics. The latter are especially recommended under TIRF conditions for additional blocking of excitation light. The results show clear super-resolution and flat-field illumination in the entire field of view. One could easily increase the field of view by changing the demagnification factor between SLM and focus plane and using a SLM with higher digital resolution. The demagnification factor is currently 0.0045 resulting in a projected SLM pixel size of 36.9 nm. This factor was chosen to increase excitation intensity for single-molecule imaging. One could increase it by a factor of 2 to still stay above Nyquist-Shannon sampling frequency. Combined with the newest generation of SLMs (e.g. the 2k 0.94" M180 FLCOS by Kopin formerly known as Forth Dimension Displays) providing 2k resolution, one could increase the field of view to approx. 100 x 100 μm^2 . Increasing the field of view results in longer readout times of the camera chip, leading to lower temporal resolution. Therefore, a field of view of 25-50 μm is often preferable for imaging fast cellular dynamics on single-cell level.